



# Equilibrium state and sensitivity of the simulated middle-to-late Eocene climate

Michiel Baatsen[1], Anna S. von der Heydt[1], Matthew Huber[2], Michael
A. Kliphuis[1], Peter K. Bijl[3], Appy Sluijs[3], and Henk A. Dijkstra[1]

[1]Institute for Marine and Atmospheric Research, Department of Physics, Utrecht University,
Princetonplein 5, 3584CC Utrecht, Netherlands

[2]Purdue University, 610 Purdue Mall, West Lafayette, IN, 47906 USA

[3]Marine Palynology and Paleoceanography, Laboratory of Palaeobotany and Palynology,
Department of Earth Sciences, Utrecht University, Princetonlaan 8a, 3584 CB Utrecht, the
Netherlands

*Correspondence to:* Michiel Baatsen (m.l.j.baatsen@uu.nl)

**Abstract.** While the early Eocene has been considered in many modelling studies, detailed simula-
tions of the middle and late Eocene climate are currently scarce. To understand Antarctic glaciation
at the Eocene-Oligocene Transition ($\sim$34Ma) as well as middle Eocene warmth, it is vital to have
an adequate reconstruction of the middle-to-late Eocene climate. Here, we present a set of high res-

olution coupled climate simulations using the Community Earth System Model (CESM) version 1.
Two middle-to-late Eocene cases are considered with new detailed 38Ma geographical boundary
conditions with a different radiative forcing. With 4x pre-industrial concentrations of $CO_2$ (i.e. 1120
ppm) and $CH_4$ ($\sim$2700 ppb), the equilibrium sea surface temperatures correspond well to avail-
able late middle Eocene (42-38 Ma) proxies. Being generally cooler, the simulated climate with 2x

pre-industrial values is a good analog for that of the late Eocene (38-34 Ma). Deep water forma-
tion occurs in the South Pacific Ocean, while the North Atlantic is strongly stratified and virtually
stagnant. A shallow and weak circumpolar current is present in the Southern Ocean with only mi-
nor effects on southward oceanic heat transport within wind-driven gyres. Terrestrial temperature
proxies, although limited in coverage, also indicate that the results presented here are realistic. The

reconstructed 38Ma climate has a reduced equator-to-pole temperature gradient and a more sym-
metric meridional heat distribution compared to the pre-industrial reference. Climate sensitivity is
similar ($\sim$0.7°C/Wm$^2$) to that of the present-day climate ($\sim$0.8°C/Wm$^2$; 3°C per $CO_2$ doubling),
with significant polar amplification despite very limited sea ice and snow cover. High latitudes are
mainly kept warm by albedo and cloud feedbacks in combination with global changes in geography

and the absence of polar ice sheets. The integrated effect of geography, vegetation and ice accounts
for a 6-7°C offset between pre-industrial and 38Ma Eocene boundary conditions. These 38Ma sim-



ulations effectively show that a realistic middle-to-late Eocene climate can be reconstructed without
the need for greenhouse gas concentrations much higher than proxy estimates. The general circu-
lation and radiative budget allow for mild high-latitude regions and little to no snow and ice cover,
without making equatorial regions extremely warm.

## 1    Introduction

The Eocene-Oligocene Transition (EOT) is one of the most dramatic climate transitions of the Ceno-
zoic, thought to be associated with the formation of a continental-scale ice sheet on Antarctica
(Zachos et al., 1994; Coxall et al., 2005; Lear et al., 2008). A possible cause for the inception of
ice is a long-term decline of greenhouse gas concentrations through the middle Eocene, eventually
crossing a threshold for glaciation (DeConto and Pollard, 2003; DeConto et al., 2008; Gasson et al.,
2014). Following the early Eocene (∼50Ma), a general cooling levelled off to a plateau in the middle
Eocene (43-42Ma) and eventually reversed into a warming until ∼40Ma (Zachos et al., 2001, 2008;
Bijl et al., 2009). At the Middle Eocene Climatic Optimum (MECO; Bohaty and Zachos 2003;
Sluijs et al. 2013), conditions returned to values close to those seen in the early Eocene (Zachos
et al., 2008; Cramer et al., 2009; Bijl et al., 2010, 2013) and quickly cooled down again into the late
Eocene (∼38Ma). This was followed by a cooling event at ∼37.3Ma characterised by the Priabonian
Oxygen isotope Maximum (PrOM, Scher et al. 2014). Significant swings in global temperature thus
occurred prior to the EOT. Yet, conditions only allowed the growth of a continental-scale Antarctic
ice sheet after 34Ma, although indications for significant ice volume in the late Eocene have been
found (Scher et al., 2014; Passchier et al., 2017; Carter et al., 2017). It remains a question to what ex-
tent continental geometry (e.g. opening of Southern Ocean Gateways) next to gradual shifts in both
the atmospheric and oceanic circulation, was a driver to both regional and global climate change
during the Eocene (Bijl et al., 2013; Bosboom et al., 2014; Sijp et al., 2014, 2016). Both the timing
and effects of Southern Ocean Gateways opening during the Eocene and Oligocene remain uncertain
and are not necessarily related to the EOT (Stickley et al., 2004; Lagabrielle et al., 2009).

Prior to the EOT, not only the exceptionally warm early Eocene (Greenwood and Wing, 1995; Bijl
et al., 2009; Huber and Caballero, 2011), but also the later part of the Eocene were characterised
by a low equator-to-pole temperature gradient (Bijl et al., 2009; Hollis et al., 2012; Douglas et al.,
2014; Evans et al., 2018). In general, the Eocene greenhouse climate has proven challenging to simu-
late adequately with climate models (Huber and Sloan, 2001; Huber and Caballero, 2011). Previous
model simulations needed very high radiative forcing in order to reproduce high-latitude warmth
but at the expense of equatorial temperatures being significantly higher than indicated by proxy data
(Huber and Caballero, 2011; Lunt et al., 2012). In more recent studies, consensus between models
and proxy data is growing as equatorial temperature estimates have risen (Pearson et al., 2007; In-



glis et al., 2015; Evans et al., 2018). Meanwhile, estimates of high-latitude temperatures are possibly biased towards warm season conditions (Sluijs et al., 2006; Bijl et al., 2009; Schouten et al., 2013). Furthermore, global climate models are continuously being improved by including more processes

(especially cloud properties, e.g. Abbot et al. 2009; Kiehl and Shields 2013) and using a higher spatial resolution with better resolved palaeogeographies (Baatsen et al., 2016; Lunt et al., 2016; Hutchinson et al., 2018).

With these improvements model simulations can provide a much higher level of detail, yet it remains unclear whether the simulated climate can match the one reconstructed from proxies. Most recent

modelling studies focussed on reconstructing either early Eocene extreme (Huber and Caballero, 2011; Lunt et al., 2012; Herold et al., 2014) or the latest Eocene - early Oligocene (Hill et al., 2013; Ladant et al., 2014; Kennedy et al., 2015) conditions. Both proxies and especially model results are relatively scarce for the middle-late Eocene considered here (Bartonian and Priabonian $\sim$41.2- 33.9Ma). An overview of many palaeoclimate modelling studies involving the Eocene is given by

Gasson et al. (2014), where the atmospheric fields are used as drivers for an ice sheet model to study the Eocene-Oligocene Transition (EOT, $\sim$34Ma). In these studies, either early Eocene boundary conditions were applied (Lunt et al., 2010b; Winguth et al., 2010; Huber and Caballero, 2011; Goldner et al., 2013) or global climate models of reduced complexity (DeConto et al., 2008; Heinemann et al., 2009; Sagoo et al., 2013) were used. Finally, a comprehensive model study of four time slices

covering the Eocene (including Bartonian/Priabonian) is presented in Inglis et al. (2015) and Lunt et al. (2016), using a low resolution version of the HadCM3 model with a relatively short spin-up of 1422 years.

In order to better understand the climatic changes that took place before the EOT, more adequate

simulations of the global climate system during the middle-late Eocene are needed. This period is interesting for climate modelling as, at the same geographic boundary conditions, it exhibits both early Eocene-like warmth (MECO) as well as conditions cool enough for possible (limited) precursor glaciations of Antarctica (PrOM). Here, we present the results of a set of coupled atmosphere-ocean simulations with the Community Earth System Model (CESM) using 38Ma boundary condi-

tions (Baatsen et al., 2016). The newly reconstructed geographic boundary conditions are at higher spatial resolution than previous ones (e.g. Sewall et al. 2000), and representative of the Bartonian (41.2-37.8 Ma) and Priabonian (37.8-33.9 Ma). Previously used geographic reconstructions (Herold et al., 2014; Lunt et al., 2017) were more representative for the early Eocene. The results presented in this paper include two cases, one with low (2x pre-industrial $CO_2$ and $CH_4$) and another with

high (4x pre-industrial $CO_2$ and $CH_4$) greenhouse gas concentrations. These $CO_2$ values (560 and 1120pp, respectively) cover most of the range in proxy estimates for the Bartonian and Priabonian ($\sim$500-1200ppm; Beerling and Royer 2011; Anagnostou et al. 2016 and latest estimates from http://www.p-co2.org) and allow for an analysis of equilibrium climate sensitivity. Both $CO_2$ and



$CH_4$ are increased with the same factor, in order to represent middle-late Eocene estimates (Beerling

et al., 2009, 2011; Herold et al., 2014).

The model set-up and spin-up procedure for the CESM simulations are first explained in section
2. The equilibrium climate of each simulation, as well as a comparison to proxy data will then be
presented in section 3 (3.1: Ocean - 3.2: Atmosphere). This is followed by an analysis of equilib-

rium climate sensitivity derived from the different simulations in section 3.3, with a focus on the
main changes involved in the radiative balance. Finally, the main results will be summarised and
discussed in section 4.

## 2    Model Set-up

For this study, version 1.0.5 of CESM (Gent et al., 2011) is used with the 38Ma geography re-
construction from Baatsen et al. (2016). The POP ocean component runs on a $\sim1°$ (384 x 320)
curvilinear grid with its northern pole located in central Greenland, and 60 vertical layers. The thick-
ness of these layers depends on depth, being 10m in the upper 20 layers and gradually increasing to
500m at a maximum depth of 5500m. The ocean is coupled to the CAM4 atmosphere component

that uses a $\sim2°$ (90 x 144) finite volume grid and 26 vertical layers extending upwards to 2hPa. Veg-
etation is fixed; generally tropical or subtropical in low-to-middle latitude regions (savannah/shrubs
in sub-tropical continental interiors) and mixed forests further poleward. Aerosol concentrations are
fixed and determined by running the atmospheric component of the model for 50 years as a bulk
aerosol model (BAM; Kiehl et al. 2000). The 38Ma simulations have 2x and 4x pre-industrial levels

(280ppm; 671ppb) of both $CO_2$ and $CH_4$, referred to as 2x and 4x PIC. Using the estimated radia-
tive forcing from Etminan et al. (2016), our simulations are comparable to $2.15\times$ ($\sim600$ppm) and
$4.69\times$ ($\sim1300$ppm) pre-industrial $CO_2$-equivalent for 2x and 4x PIC, respectively. For reference, a
pre-industrial simulation is performed using the same model version with 1x PIC and present-day
boundary conditions for geography and vegetation. A short overview of the main characteristics of

each simulation is given in Table 1.

Both 38Ma model simulations start from the same initial conditions: a stagnant ocean with a hor-
izontally homogeneous temperature distribution. The initial ocean temperature decreases linearly

with depth, from 15°C at the surface to 9°C at the bottom. The pre-industrial reference is initialised
using temperature and salinity fields from the PHC2 dataset (Steele et al., 2001). A long spin-up
(see Table 1) is performed to allow the deep ocean to equilibrate sufficiently. An overview of ab-
solute ($\Delta T$, $\Delta S$) and normalised ($\Delta T/T$, $\Delta S/S$) drifts over the last 200 model years is given in





| Simulation / Properties | Pre-industrial | 2x PIC 38Ma | 4x PIC 38Ma |
|---|---|---|---|
| Geography | Present day | 38Ma Paleomag (Baatsen et al., 2016) | |
| Vegetation | Present day | 38Ma; plant functional types fixed | |
| Aerosols | Pre-industrial | from 50-year BAM run | |
| $CO_2$ | 280ppm | 560ppm | 1120ppm |
| $CH_4$ | 671ppb | 1342ppb | 2684ppb |
| Spin-up | 3000 years | 3000 years | 4000 years |

**Table 1.** Overview of characteristics for all CESM 1.0.5 simulations that were performed (BAM: bulk aerosol model).

Table 2 for each spin-up. Drifts are generally $\sim 10^{-4}$ K/year for global mean, volume weighted av-
erage ocean temperature. Similarly, for globally averaged salinity drifts at the end of the spin-up are
$\sim 10^{-6}$ psu/year.

| Simulation / Measure | Pre-industrial | 2x PIC 38Ma | 4x PIC 38Ma |
|---|---|---|---|
| $\Delta T$ (K/year) | $-1.2 \cdot 10^{-4}$ | $5.7 \cdot 10^{-5}$ | $2.6 \cdot 10^{-4}$ |
| $\Delta S$ (psu/year) | $1.9 \cdot 10^{-6}$ | $1.7 \cdot 10^{-7}$ | $-1.0 \cdot 10^{-6}$ |
| $\Delta T/T$ (/year) | $-4.3 \cdot 10^{-7}$ | $2.0 \cdot 10^{-7}$ | $9.1 \cdot 10^{-7}$ |
| $\Delta S/S$ (/year) | $5.4 \cdot 10^{-8}$ | $4.9 \cdot 10^{-9}$ | $-3.0 \cdot 10^{-8}$ |

**Table 2.** Overview of drifts in global mean ocean temperature and salinity over the last 200 model years for all three CESM spin-up simulations. Normalised drifts $\Delta T/T$ and $\Delta S/S$, with $T$ temperature in Kelvin and $S$ salinity in psu, are also shown for each case using the same model period.

Time series of the horizontally averaged upper ($<$1000m), deep ($>$2000m) and full depth ocean
temperatures are shown for both 38Ma and pre-industrial spin-up runs in Figure 1a-b. Starting from
present-day initial conditions, the pre-industrial simulation cools down by about 0.5°C globally. In
addition to globally averaged temperature, the average is also shown for the Pacific and Atlantic
basins separately. The (38Ma) 2x PIC simulation is seen to equilibrate much faster than the 4x PIC
one, probably because the deep ocean equilibrium temperature is close to that of the initial state
($\sim$9°C). With higher greenhouse gas concentrations, the ocean experiences additional heating at the
surface, causing it to become more strongly stratified which consequently reduces vertical mixing
into the deep ocean. This reduced mixing in the 4xPIC case causes the global mean deep ocean
temperature to still increase about 0.1°C over the last 500 years. As expected, changes in the up-





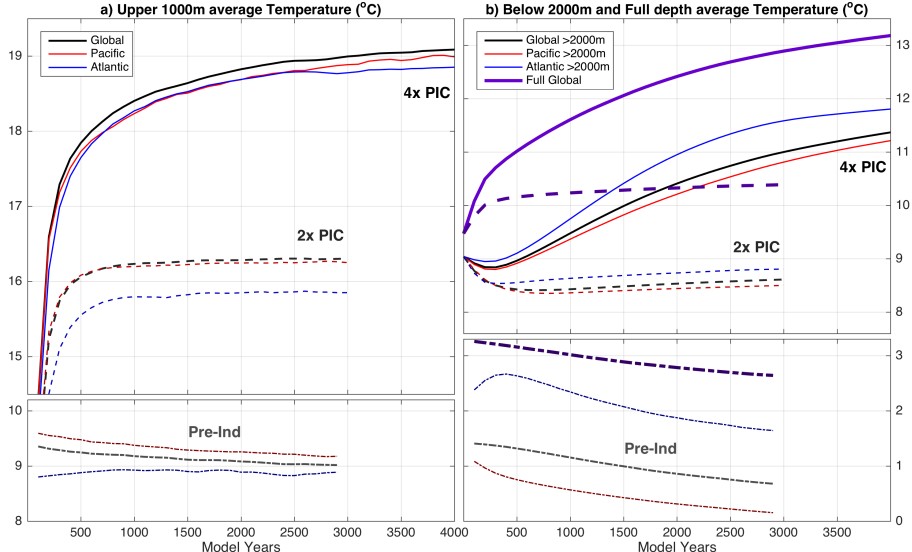

**Figure 1.** Time series of **a)** upper 1000m and **b)** below 2000m volume-weighted average temperature for Pre-Industrial (dotted), 38Ma 2x (dashed) and 4x (solid) PIC spin-up simulations. Averages are shown globally (black) and for both the Pacific (red) and Atlantic (blue) ocean basins separately. Full depth global average temperature is also shown for each run with thick purple lines

per 1000m temperatures settle more quickly and are small (∼0.1°C) for the last 2000 model years

of each simulation. In addition, global patterns of ideal age tracers (a measure for oceanic venti-
lation timescales; Thiele and Sarmiento 1990; England 1995) are observed to equilibrate over the
last 1000 years of each simulation (Figure 2) as well as meridional overturning strength and gate-
way transports (shown in Section 3.1). Average values of ideal age still have considerable trends
because of further ageing in stagnant deep ocean regions (Figure 2a) and upwelling of older water

masses into the upper ocean (Figure 2b). Such changes will likely continue for many thousands of
years, but the associated circulation pattern has equilibrated over the last 200 years of the simulation.

## 3 Results: middle-to-late Eocene equilibrium climate

### 3.1 Ocean

For each simulation, averages are made using over last 50 model years to represent the equilibrium
climate. The resulting annual mean ocean state for the 4x PIC 38Ma climatology is visualised by
the set of fields shown in Figure 3. First of all, sea surface temperatures are quite warm at low
latitudes (23°S-23°N average of 33.9°C, locally >36°C) but also mild in the high-latitude regions





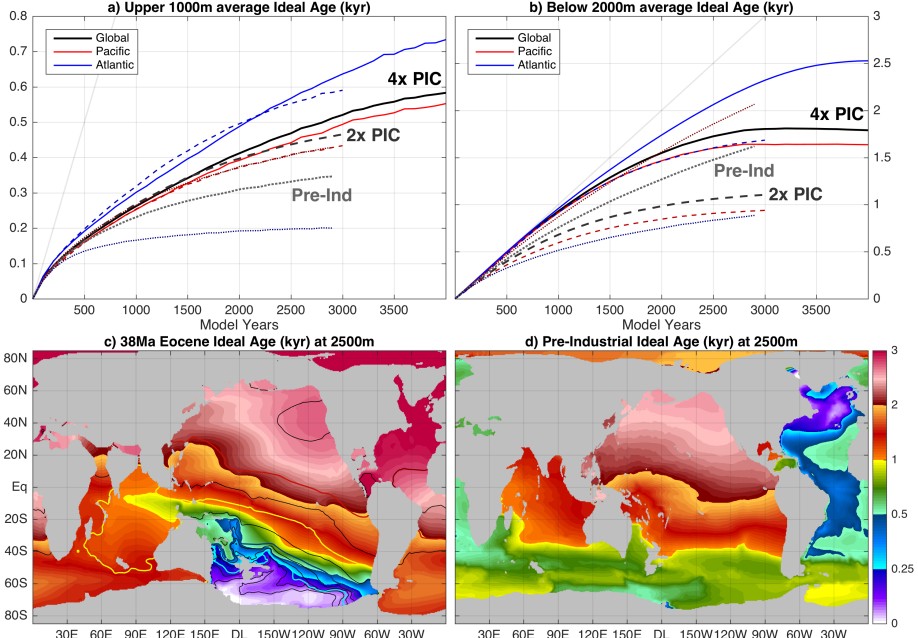

**Figure 2.** Time series of **a)** upper 1000m and **b)** below 2000m volume-weighted average ideal age tracers for Pre-Industrial (dotted), 38Ma 2x (dashed) and 4x (solid) PIC spin-up simulations. Averages are shown globally (black) and for both the Pacific (red) and Atlantic (blue) ocean basins separately. **c)** 38Ma and **d)** pre-industrial horizontal distributions of ideal age tracers at 2500m depth for the end of each model simulation. Color shading in c) shows the ideal age of 4x PIC, while contours do so for the 2x PIC case (black line every 100 years up to 500 and every 250 years above, coloured contours use the color bar shown to the right).

(Figure 3a). Mainly the Southern Ocean and especially its Pacific part is characterised by annual

mean temperatures of 10-20°C, while those of the Arctic Ocean are generally 6-8°C. The former Tethyan (now Indo-Pacific) Ocean still acts as one system in the equatorial region, with an expansive warm pool and cold tongue across its western and eastern part, respectively. Mild temperatures in the sub-polar South Pacific are accompanied by relatively high salinities around 35psu (Figure 3b). Higher salinities are present in low latitude evaporative regions around the world, in contrast to

the much fresher Arctic Ocean (∼20psu). Low temperatures and salinities in the Arctic Ocean are related to the basin being mostly isolated, apart from the shallow Turgai Strait (<100m depth) and connections to the North Atlantic Ocean (<1000m depth), which has low salinities as well. These surface temperature and salinity patterns are reflected by upper ocean potential density (Figure 3d), with high values throughout the Southern Ocean and much lower densities and thus more stable

waters in the North Atlantic and Arctic Oceans. The latter is consistent with proxy indications of a stratified North Atlantic Ocean (Coxall et al., 2018). Deep water formation occurs only in the South





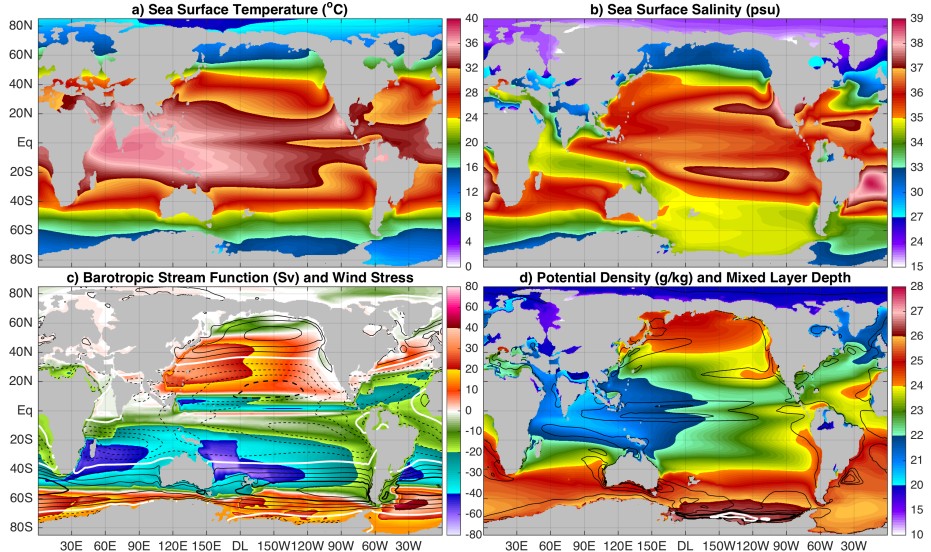

**Figure 3.** Annual mean **a)** sea surface temperature and **b)** salinity, **c)** barotropic stream function (positive for clockwise flow) and zonal wind stress (contours every $2.5 \cdot 10^{-2}$Pa, thick lines every $1 \cdot 10^{-2}$Pa; solid positive and dashed negative, thick white line at 0Pa), and **d)** upper 200m mean potential density and mixed layer depth (contours every 50m, thick lines every 200m, thick white line at 500m) for the 4x PIC simulation.

Pacific in winter, but low stratification suggests that this may take place virtually anywhere around the Antarctic continental slope. The only location in the Northern Hemisphere where deep water formation could potentially take place consists of the weakly stratified high latitudes of the North

Pacific, but it does not occur in these simulations. North Pacific deep water formation is seen in various Eocene simulations (Thomas et al., 2014; Baatsen et al., 2018; Hutchinson et al., 2018) and also suggested by proxy data (Hague et al., 2012).

Most of the global circulation is dominated by sub-tropical and sub-polar gyres (Figure 3c), with the strongest cell in the South Indo-Pacific. A frontal zone is present in the Southern Ocean, separating

cyclonic (sub-tropical) and anticylonic (sub-polar) gyres. This front coincides with an Antarctic Circumpolar Current (ACC) that is, however, still strongly restricted in its path and depth (200-500m). Strikingly, the front is located at 55-60°S, which is 10° poleward of where it is today. The region of maximum zonal wind stress coincides with this front, showing that it is truly fixed by the latitudes where winds are the least obstructed by continents. Due to the more southerly position of Australia in

the Eocene, the temperature front separating warm Pacific from colder Antarctic waters also moves further southward. Interestingly, this allows low-latitude derived waters to extend much further south helping to explain high temperatures in many Southern Ocean regions, especially the Southwest Pacific in summer (Bijl et al., 2009; Douglas et al., 2014). On the other hand, a strong eastward flow



through the Tasmanian Gateway limits the strength and extent of the Antarctic counter current of

which the influence is seen at most nearby sites throughout the Eocene Stickley et al. (2004); Huber et al. (2004); Bijl et al. (2013). The circulation pattern and associated temperature/salinity fields found here generally correspond well with those shown by Hutchinson et al. (2018), who use the GFDL model with the same geographic boundary conditions.


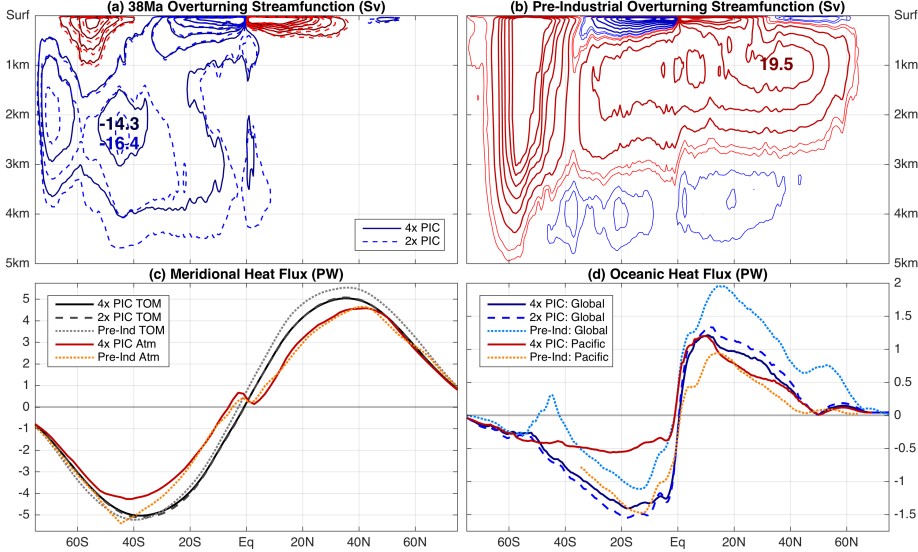

**Figure 4.** Global oceanic meridional overturning stream function, averaged for the last 50 model years of the **a)** 4x (solid) and 2x (dashed) PIC Eocene simulation, and **b)** pre-industrial reference. Red contours show positive and blue negative values, drawn every 5Sv (thin lines show 1 and 2Sv in b) with the absolute maximum of the strongest cell (with sign) below 1000m specified. Top of model required total meridional heat flux is shown in **c)** for 4x PIC (solid black), 2x PIC (dashed dark grey) and pre-industrial (dotted grey) with the corresponding atmospheric fluxes in red (4x PIC) and dotted orange (pre-industrial). **d)** Total integrated meridional heat fluxes in the ocean, globally (blue) and Pacific-only (red). Note that all horizontal (latitude) scales cover the range [75°S, 75°N] with latitude increasing from left to right, for better comparison with oceanic fields.

Despite differences in spin-up time, patterns of the equilibrium ocean circulation state are generally similar for the 2x and 4x PIC simulations. The annual mean, global mean sea surface temperature (SST) is 28.4°C in the 4x PIC case versus 25.8°C in the 2x PIC one (the pre-industrial model value is 18.4°C). Temperature differences between both 38Ma cases of 2.5-3°C are also seen in the upper

and deep ocean in Figure 1, and are close to the global mean SST change of 2.6°C. Without significant continental ice sheets, a similar 2-4°C cooling would be expected between the middle and late Eocene corresponding to a benthic $\delta^{18}$O difference of 0.5-1‰ (Zachos et al., 2001, 2008; Cramer



et al., 2009). In our Eocene simulations, the globally averaged deep-sea temperature (below 2000m) reaches ~11.5°C in the 4x PIC case (and ~8.5°C in the 2x PIC case), which is much warmer than

for the pre-industrial ocean. In therms of an oxygen isotope ($\delta^{18}$O) signal, this corresponds to a ~2.8‰ difference due to deep sea warming from the pre-industrial to the Eocene 4x PIC climate and ~2‰ from pre-industrial to 2xPIC Eocene. A more elaborate description of global and regional temperature change patterns associated with climate sensitivity and geography is given in section 3.3.

The globally integrated meridional overturning stream function is similar in pattern and extent (Figure 4a) for both Eocene simulations. Shallow wind driven cells are evident around the equator and a deep overturning cell is present in the Southern Hemisphere, which is slightly stronger in the 2x PIC case. The effect of upwelling and a shallow positive overturning cell, linked to a proto-ACC, can be seen at around 55°S in both cases. A much stronger and deep (northern) overturning cell is

seen in the pre-industrial reference, with upwelling near the ACC location. Note that the differences in position of the upwelling cells between Eocene and pre-industrial circulation is smaller than those seen for the surface polar front in Figure 3.

An otherwise predominantly wind-driven gyre circulation is reflected by symmetric meridional oceanic heat fluxes into both hemispheres (Figures 4d). This is in contrast to the pre-industrial situa-

tion, where approximately a 1PW maximum difference between hemispheres is observed making the Northern Hemisphere relatively warm (Trenberth and Caron, 2001). A major part of the heat flux is generated in the Pacific Ocean, with the exception of the southern low latitudes. A large contribution arises from both the South Indian and South Atlantic sub-tropical gyres. A subtle but important difference between hemispheres is seen at high latitudes, where about 0.5PW is transported southward

at 45°S while the heat transport is close to zero at 45°N. This difference can be partly explained by the presence of a deep meridional overturning cell (with negative sign) in the South Pacific, pulling warm waters into the southern high latitudes. In addition, land masses obstructing zonal flow at middle-high latitudes (more strongly than at present day) allow for stronger meridional contributions in gyres. Top of model net shortwave and longwave fluxes are integrated to obtain the required

compensating meridional heat flux (Figure 4c), which does not differ much between both Eocene and the pre-industrial cases.

The ocean heat transport patterns in our 38Ma Eocene simulations generally agree with those seen in previous model studies. In situations with no (or a restricted) ACC, increased southward heat transport is found in the Southern Hemisphere and attributed to both changes in the horizontal gyre

circulation (in particular sub-polar gyres; Huber et al. 2004; Huber and Nof 2006; Sijp et al. 2011) and more directly to the meridional overturning circulation (Toggweiler and Bjornsson, 2000; Sijp and England, 2004; Sijp et al., 2009). These differences with respect to the present (pre-ndustrial) ocean circulation have been referred to as the Drake Passage effect Toggweiler and Bjornsson (2000), which has been shown to also exist (with different strength) in model simulations resolving meso-





scale ocean eddies Viebahn et al. (2016).

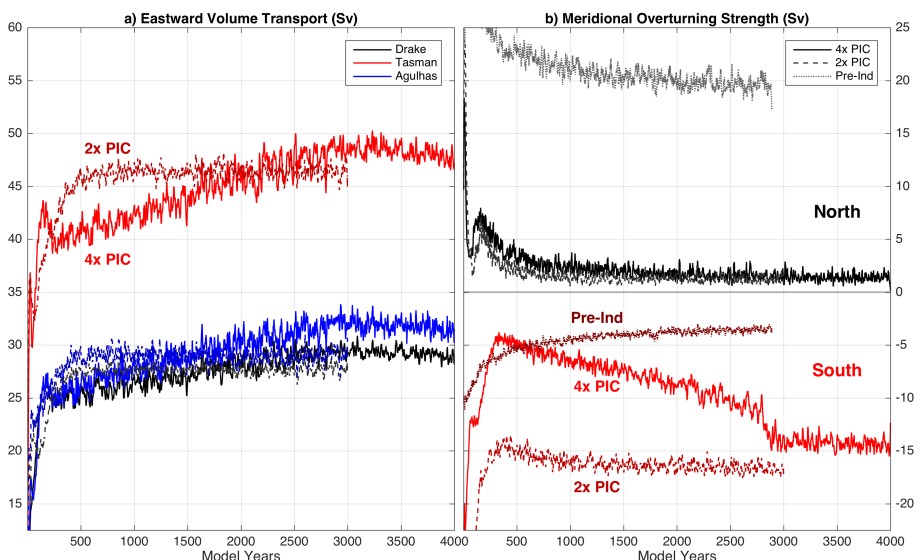

**Figure 5.** Time series of **a)** Southern Ocean Gateway volume transports and **b)** maximum global meridional overturning for both 2x (dashed), 4x (solid) PIC Eocene and pre-industrial (dotted) spin-up simulations. Transports through three north-south transects in the Southern Ocean are considered: Drake Passage (black, 65°W), Tasmanian Gateway (red, 150°E) and Agulhas (blue, 25°E), with positive values indicating eastward flow.

Subtle differences in spin-up between both 38Ma Eocene simulations are highlighted by time series of gateway transports and global meridional overturning (Figure 5). Early in the 4x PIC run the

southern overturning cell weakens considerably (Figure 5b), as the ocean is heated from above and becomes stably stratified. Through diffusion, the deep ocean gradually warms up facilitating deep convection in the Southern Ocean. Only after 2700 model years, a deep southern overturning cell develops with a similar strength to that of the 2x PIC run. Volume transports through the Drake Passage, Tasmanian Gateway and Agulhas transect are shown in Figure 5a. Induced by the Southern Ocean

(and especially South Pacific) meridional density gradient, zonal transports through the considered transects are connected to the overturning strength. Even after using a 10-year running mean, the smoothed time series still show considerable variability involving Rossby waves and multi-decadal scale processes. Additionally, transport strengths are relatively high considering the limited depth and width of both the Drake Passage and Tasmanian Gateway. With 45-50Sv of volume transport,

the flow through the Tasmanian Gateway is almost one third of its present-day equivalent (∼180Sv in the pre-industrial reference). A strong forcing is provided by the meridional density gradient in





the South Pacific, pushing water through the Southern Ocean Gateways. Continental geometry, being consistent with proxy-based reconstructions (Stickley et al., 2004), does not yet allow the development of a deep ACC.


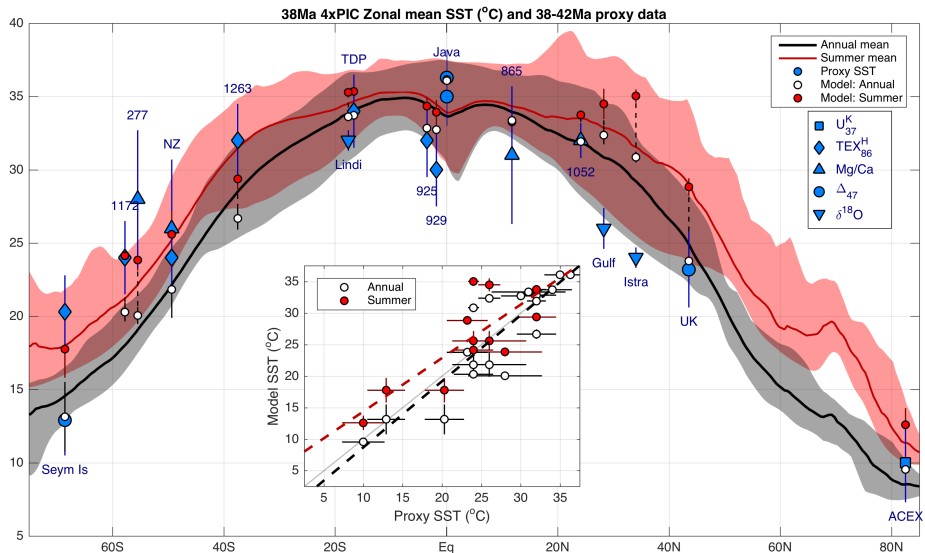

**Figure 6.** Zonal mean, annual mean (black) and summer mean (red) sea surface temperature (SST) with shaded regions showing minimum and maximum values for each latitude (4x PIC simulation). Blue markers indicate estimates from 42-38 Ma SST proxies, whereas the white (annual) and red (summer) circles depict model values at the corresponding 38Ma locations. Error bars are obtained using proxy calibration errors and the spatial variation within a $4° \times 4°$ box surrounding the corresponding location in the model. The inset shows a scatter plot comparing proxy and model SST, with dashed lines indicating a linear fit using annual mean (black) or summer mean (red; only poleward of $23°$N/S) model temperatures.

As a validation of the model output, sea surface temperatures of the 4x PIC climatology are compared to estimates from 42-38 Ma proxy records (Appendix: Table A1, data from Pearson et al. (2001, 2007); Tripati et al. (2003); Kobashi et al. (2004); Bijl et al. (2009); Liu et al. (2009); Okafor

et al. (2009); Douglas et al. (2014); Hines et al. (2017); Evans et al. (2018)). The model zonal mean, annual mean is given by the black line in Figure 6 with the (geographical) temperature range at each latitude indicated by the shaded area. A point-by-point comparison with proxy records is made, using the estimated 38Ma positions in accordance with van Hinsbergen et al. (2015) and Baatsen et al. (2016). Proxy-derived values for annual mean temperatures are indicated by blue markers with

their calibration uncertainty for different methods (UK$_{37}$, TEX$_{86}^{H}$, Mg/Ca, $\Delta_{47}$ and $\delta^{18}$O). Estimates



using UK$_{37}$ are disregarded at low latitudes as the calibration saturates at ∼28°C, those from Mg/Ca
use the constraints on sea water chemistry from Evans et al. (2018). Corresponding SST values from
the simulated climate are indicated by white and red circles for the annual and summer mean, re-
spectively. Error bars on model temperatures are obtained by taking the variance in a $4° × 4°$ box
surrounding the reconstructed proxy location, covering the uncertainty associated with the palaeo-
geographic reconstruction (http://www.paleolatitude.org).

In low-latitude regions, model results and proxies are in reasonable agreement, with model estimates
mostly in the upper range of SST estimates. Some near-equatorial sites (e.g. Atlantic ODP 925 and
929) show colder temperatures, near 30°C. This can be caused by the model's limitation in resolv-
ing the sharp equatorial upwelling zone with significantly colder waters, or a slight discrepancy in
its exact position. The existence of colder temperatures in the model is indicated by the increased
spread zonally, with <30°C near the equator. TEX$_{86}^H$ estimates from Tanzania show near-equatorial
temperatures around 34°C and are indicative of waters outside upwelling zones. Additionally, recent
clumped isotope measurements from Java suggest 35-36°C in the West Pacific warm pool, which is
in good agreement with the model.

A mixed agreement is seen at other latitudes, with model results being too warm in the northern
middle latitudes, too cold in southern middle latitudes and good at high latitudes. The large discrep-
ancy with proxy estimates seen in the Gulf of Mexico and Neotethys (Mediterranean) stands out.
While it seems unlikely that these regions would be colder than they are today, the model possibly
also underestimates strong seasonal cooling in shallow coastal waters and other local effects near
the coastline. The complex palaeogeography at both locations is not sufficiently represented even
in our relatively high-resolution simulations. SST estimates from other locations at similar latitude,
such as ODP 1052 on Blake Nose (east of Florida) are much higher bringing the model and proxies
into good agreement. A better match is generally found at middle and high latitude regions (apart
from the Arctic) when summertime temperatures are considered. Since SST proxies are based on
past living organisms, it is possible that at higher latitudes these proxies have a bias towards the
warm season as their activity and sedimentation are directly or indirectly affected by the amount of
sunlight (Sluijs et al., 2006, 2008; Bijl et al., 2009; Hollis et al., 2012; Schouten et al., 2013).

A comparison of 2x PIC model SST with 38-34 Ma proxy estimates shows a similar result (Ap-
pendix: Table A2 and Figure A1). Good agreement between proxy and model results exists in equa-
torial regions, with again a large spread in proxies and generally a better match with summer temper-
atures at higher latitudes. Assuming that temperature proxies represent the annual mean in equatorial
waters but are skewed towards the warm season at middle and high latitudes, the model does seem
to reconstruct the middle-to-late Eocene (42-34 Ma) temperature distribution quite well.






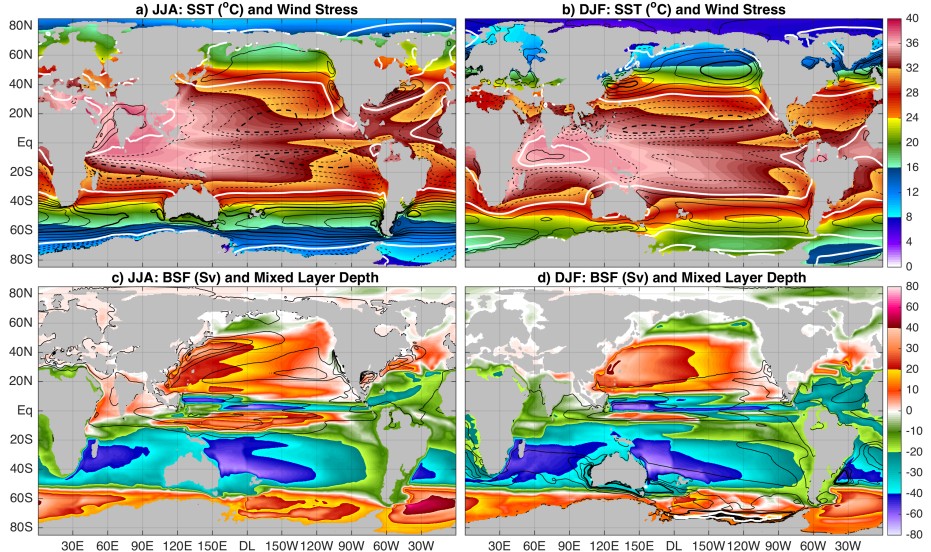

**Figure 7.** 4x PIC simulation **a)** June-July-August and **b)** December-January-February averaged sea surface temperature and zonal wind stress (contours every $2 \cdot 10^{-2}$ N/m$^2$ and thick white lines every $5 \cdot 10^{-2}$ N/m$^2$). **c)** and **d)** similar to a) and b), but for the barotropic stream function and maximum mixed layer depth (contours every 100m and thick white lines every 500m).

In addition to the annual mean fields shown in Figure 3, winter and summer averaged SST, wind stress, barotropic stream function and mixed layer depth are presented in Figure 7. Sea surface salinity and upper 200m potential density are not considered here as they show little seasonality. At
southern high latitude regions, SST seasonality is strong with winter temperatures of mostly 6-12°C rising to 16-24°C in summer. A combination of solar heating and gyre circulations causes the coldest summer-time temperatures in the Southern Hemisphere to occur at the centre of sub-polar gyres, rather than near the Antarctic coastline (zonal mean temperatures are also seen to increase slightly beyond 65°S in Figure 6). Not surprisingly, the South Pacific holds the warmest high latitude waters
as it exhibits additional southward heat transport in the meridional overturning circulation.
Considering the barotropic stream function, warm waters in the Southwest Pacific are carried southward through the East Australia Current (EAC) down to 55°S where the EAC meets the proto-Leewin current from the west, and northward in the Ross Gyre further poleward. This is mostly consistent with the picture for the middle Eocene given by Bijl et al. (2009), but shifted southward (with a much
smaller Ross Gyre) compared to Huber et al. (2004). In contrast, seasonality outside near-equatorial regions is weakest in the Arctic Ocean (related to persistent low cloud cover). Very high summer temperatures stand out in the Mediterranean/Neotethys, caused by heat transport from the Indian Ocean warm pool in combination with strong insolation. Further north, the Turgai Straight (55°N,





60°E) separates the Arctic from sub-tropical water masses, with strong seasonality due to shallow

waters. The barotropic stream function and zonal wind stress show similar seasonal cycles (as the gyres are mainly wind-driven), with strong (weak) zonal flow and gyres in winter (summer). This can be directly related to changes in the meridional temperature gradient, which is stronger in winter. Finally, the effect of northward flowing bottom waters (restricted by bathymetry) can be seen in the South Pacific sub-polar gyre. This feature is strongest in winter, which is the time when deep water

formation is taking place as indicated by the maximum mixed layer depth.

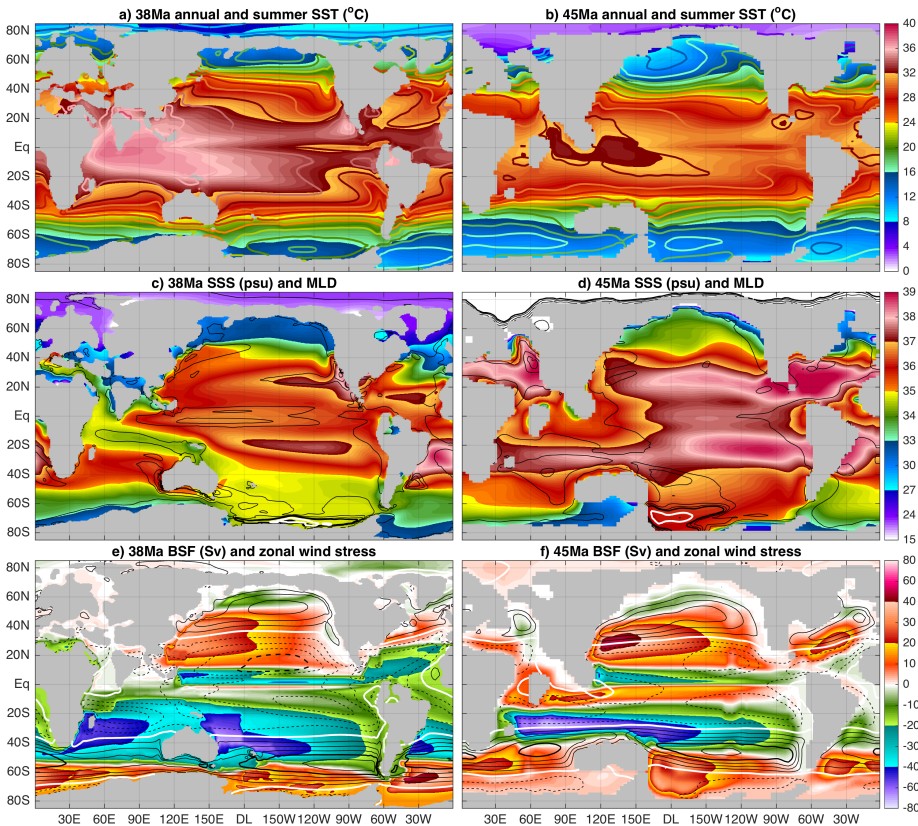

**Figure 8.** Annual mean (shading) and summer (contours every 2°C) sea surface temperatures for the **a)** 38Ma 4x PIC simulation presented here and **b)** results from Goldner et al. (2014) (GH14) using a 45Ma gegraphy and 4x pre-industrial $CO_2$. A similar comparison is made for **c-d)** sea surface salinity (psu) and mixed layer depth (contours) as well as **e-f)** barotropic stream function (shading) and zonal wind stress (contours). All color scales and contour intervals use the same conventions as those in Figures 3 and 7.



To put the results presented here into perspective, the 4x PIC case is compared to previous model simulations from Goldner et al. (2014) (hereafter: GH14) in Figure 8. The latter use a 45Ma hot spot referenced reconstruction for their geographical boundary conditions and 4x pre-industrial levels of
$CO_2$ (i.e. 1120ppm). The main difference between both studies is the horizontal resolution; $\sim1°$ and $\sim2°$ in our 38Ma simulations versus $\sim3°$ and $\sim4°$ in GH14 for the ocean and atmosphere grids, respectively. While the 38Ma 4x PIC case also has a quadrupling of $CH_4$ and the 45Ma one does not, the difference in global temperature is much larger than would be expected based on radiative forcing alone. The grid and resolution used here for the atmospheric component (CAM4) are shown
to both increase the sensitivity to a doubling of $CO_2$ by Bitz et al. (2012). Aditionally, using the finite volume instead of a spectral grid greatly influences cloud radiative forcing and causes a warming compared to GH14. Indeed, similar global temperatures are found under comparable radiative forcing by Hutchinson et al. (2018), using a lower resolution but comparable finite difference dynamical core in the Atmospheric component. Finally, a warm initialisation of the deep ocean, general
circulation changes and aerosol changes may add further to temperatures differences.

The most prominent differences in geography are the representation of Antarctica, the Tasmanian Gateway opening and the position of India. Especially the Southern Ocean shows large differences in circulation and the resulting temperatures, mainly related to the different geography. The formation of a proto-ACC and extent of the East Australia current act to shift the polar front in the South
Pacific southward for the 38Ma case, while the opposite happens for 45Ma. These changes can be linked directly to the continental configuration and associated shifts in zonal wind stress (maximum at 55°S versus 45°S). The more southerly position of the polar front can explain the heterogeneity in the Southern Ocean; a difference of 6-8°C is seen between Tasmania and the tip of the Antarctic Peninsula in agreement with proxy indications from Douglas et al. (2014). Generally, western
boundary currents (e.g. Kuroshio, Agulhas, East Australia Current) and the effects of bottom topography are more pronounced in the 38Ma results. Finally, an issue in the 45Ma results with very low (negative) salinities in the Arctic, although having seemingly little impact on the general circulation, is mostly resolved (lowest salinities down to $\sim10$ psu) in the 38Ma case by having several shallow passages.


A comparison of zonal mean SST's in the 38Ma 2x PIC and 4x PIC cases, 45Ma 4x $CO_2$ from GH14 and pre-industrial reference simulation along with proxy estimates is shown in the Appendix (Figure A3). Apart from the 38Ma cases being overall warmer, the zonal mean temperature profile seems qualitatively similar for all Eocene simulations. With several indications of near-equatorial
temperatures as high as 34-36°C (Tanzania, Java and Saint Stephens Quarry), only the 38Ma 4x PIC case seems able to match those proxies while still allowing cold upwelling zones to be below 30°C. On the other hand, cooler low-latitude proxies of $\sim30$C are better matched by both the 45Ma 4x $CO_2$ and 38Ma 2x PIC results. Southern hemisphere high latitude proxies are difficult to match by




any model because of their large spread. Still, most of the lower estimates are best matched by the
annual mean SST in the 38Ma 4x PIC case while also meeting the higher estimates when consider-
ing summer maxima.

## 3.2 Atmosphere

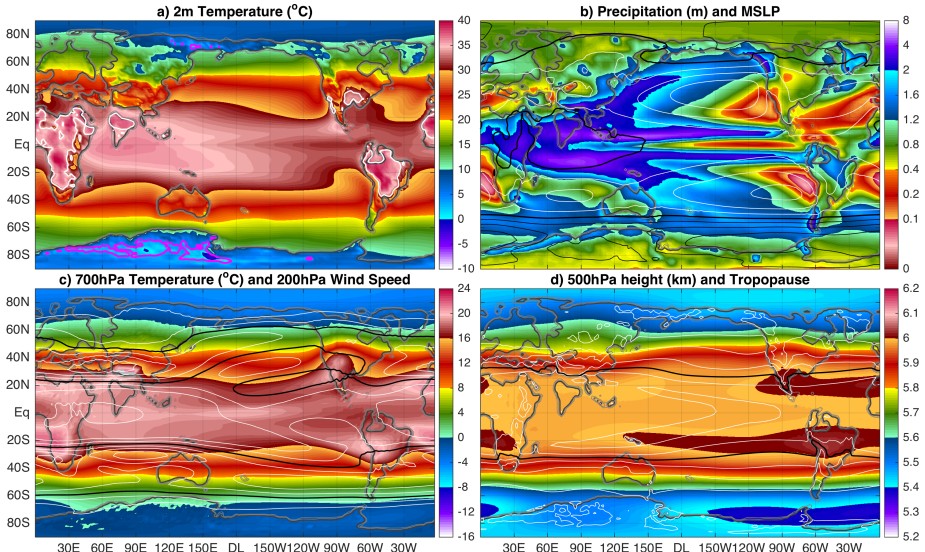

**Figure 9.** Annual mean for the 4x PIC simulation with **a)** 2m above ground level air temperature (coloured)
and average min/max temperature (contours; magenta $T_{min} < 0°$C and white $T_{max} > 40°$C), **b)** precipitation
(coloured) and mean sea level pressure (black contours every 10hPa, >1000hPa in white, thick line at 1000hPa),
**c)** 700hPa temperature (coloured) and 200hPa wind speed (white contours every 5m/s starting at 10m/s, thick
black lines every 20m/s), and **d)** 500hPa geopotential height (coloured) and dynamic tropopause height (con-
tours every 1km starting at 12km, thick black line at 16km).

Similar to oceanic fields, a climatology for the atmosphere is made from the last 50 years of the
4x PIC simulation of which the annual mean is shown in Figure 9. Again, results for the 2x PIC case
look similar apart from being 3.1°C colder, globally averaged (23.7°C versus 26.8°C). Surface air
temperatures (at 2m; Figure 9a) are generally above zero and up to 40°C in the sub-tropics, with
annual mean minimum temperatures only below freezing in East Antarctica and north-east Siberia.
Mean sea level pressure and precipitation patterns (Figure 9b) show a typical tropical trough and
sub-tropical ridges. The inter-tropical convergence zone (ITCZ) consists of 2 precipitation maxima,
extending through the Indo-Pacific basin on both sides of the equator. A pronounced double ITCZ
over the Pacific Ocean is also seen in the pre-industrial reference and a known model-related issue





(Song and Zhang, 2009; Bellucci et al., 2010). Effects of orographic lift are evident on westward facing coastlines and ridges across the middle and high latitudes.

The temperature at 700hPa (Figure 9c) highlights warm mid-level air masses in persistent continental high pressure regions in the sub-tropics. Antarctica is substantially warmer than the Arctic at this pressure level because of its elevation and continental climate. Similar to the Tibetan Plateau today, it therefore acts as an elevated heat island (Hoskins and Karoly, 1981; Ye and Wu, 1998). Wind speeds at 200hPa show maxima linked to the positions of both sub-tropical and polar jet streams. Several

stationary Rossby Waves are present, but a persistent upper level trough is evident around 90°E, where both jets coincide. Such a pattern can also be seen at present day but only in the Northern Hemisphere (located at 120°E in the pre-industrial reference), caused by the influence of the Tibetan Plateau.

Finally, the 500hPa geopotential height field (Figure 9d) shows a nearly meridionally symmetric

pattern, with the sharpest gradient in middle-latitude regions. The 500hPa surface is generally about 200m higher than in the pre-industrial reference, mainly due to warming of the air column. The current ∼200m asymmetry between southern (∼5km) and northern (∼5.2km) high latitudes does not occur in the 38Ma Eocene climate (∼5.4km for both). The corresponding tropopause height (using the WMO definition of 2°C/km) is at 17-18km in the tropics and does not fall below 11-12km at

high latitudes, being 1 and 2km higher than it is in the pre-industrial climate, respectively.

As was done for sea surface temperatures in the previous section (Figure 6), an overview of zonally averaged 2m air temperatures is given in Figure 10. Considering annual mean temperatures, a

meridionally symmetric pattern with a low equator-pole gradient stands out again. This pattern looks similar for both 38Ma simulations, being about 3°C colder in the 2x PIC case with a slight amplification towards the poles. Regardless of greenhouse gas concentrations, the pre-industrial reference temperatures are in sharp contrast with those seen in the Eocene cases. In addition to being considerably colder (global mean 2m temperature of 13.7°C), the pre-industrial climate features a stronger

equator to pole temperature gradient with high latitudes up to 40°C colder than those seen in the 38Ma 2x PIC case. This shows that the absence of ice sheets and changes in continental geometry have a significant impact on global mean temperature as well as its distribution. Despite changes in the general circulation, total meridional heat fluxes are not substantially different in the Eocene climate compared to the pre-industrial reference (Figure 4c). The reduced equator to pole temperature

gradient is thus a result of reduced heat loss at high latitudes rather than enhanced heat fluxes.

Tropical temperatures are closely tied to prevailing water masses and show little variability, both seasonally and zonally. Summer temperatures in the sub-tropics are on average comparable to those in the tropics, but with higher zonal variability. Especially around 40°N and 30°S, land masses are more abundant in the Eocene allowing for summer temperatures of up to 45°C at 4x PIC. While





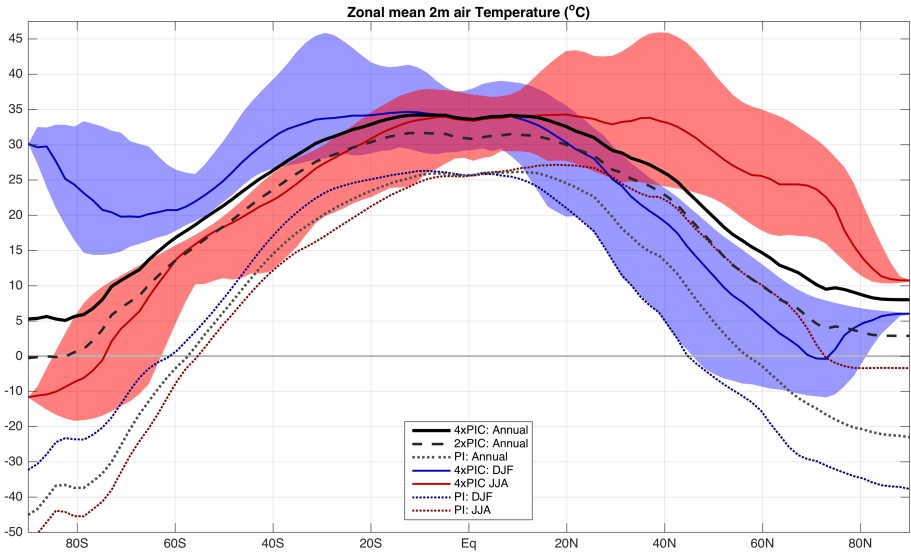

**Figure 10.** Annual mean, zonal mean 2m air temperature (black line) for the 38Ma 4x PIC (solid), 2x PIC (dashed) and Pre-industrial (dotted) equilibrium climate. Zonal mean temperatures of the 4x PIC climate, averaged for June-July-August and December-January-February are shown in red and blue, respectively. For both of the latter, the longitudinal range of temperatures is indicated by shaded areas. The same is shown for the pre-industrial seasonality, but only considering the zonal mean.

seasonality is weak at southern middle latitudes it is stronger at northern middle and high latitudes, again caused by differences in land coverage. Remarkably strong seasonality is seen on the Antarctic continent, where average winter (DJF) temperatures drop below -10°C while easily reaching 30°C in summer (JJA). The continental configuration thus causes the coldest place in winter on the Northern Hemisphere to be at ∼70°N (rather than the North Pole), while the warmest place in summer

south of ∼45°S is on Antarctica. The near meridional symmetry in temperatures is reflected by both oceanic and atmospheric heat fluxes, shown in Figure 4c-d.

     A comparison between middle-late Eocene terrestrial proxies (Appendix: Table A3, data from

Greenwood and Wing 1995; Gregory-Wodzicki 1997; Smith et al. 1998; Wolfe et al. 1998; Greenwood et al. 2004; Retallack et al. 2004; Hinojosa and Villagrán 2005; Uhl et al. 2007; Boyle et al. 2008; Schouten et al. 2008; Prothero 2008; Eldrett et al. 2009; Quan et al. 2012; Passchier et al. 2013) and 4x PIC model air temperatures is made in Figure 11. Terrestrial proxies used here consist mainly of (multiple) vegetation-based indicators using pollen, nearest living relative (NLR), leaf

physiognomy (LMA, CLAMP, ELPA and LMA; Wing and Greenwood (1993); Yang et al. (2011);





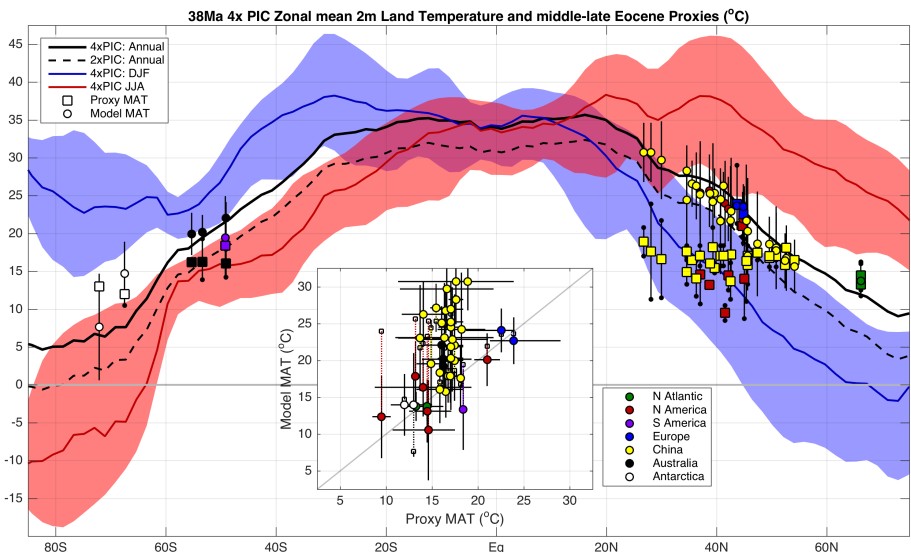

**Figure 11.** As in Figure 10 for land-only temperatures, proxy estimates (squares) and corresponding model mean annual temperature (circles). Proxy temperatures are colour coded for their region and error bars (black dots) are indicative for the spread at each site. Similar to model SST, error estimates (black lines) show the spacial variation within a $5° \times 4°$ box surrounding the corresponding location in the model. The inset shows a scatter plot comparing proxy and model air temperatures, where the latter are corrected for differences in model and reconstructed topography (uncorrected: small white squares).

Traiser et al. (2005); Kowalski and Dilcher (2003)) and MBT-CBT (Peterse et al., 2012). In contrast to Figure 10, only temperatures over land are considered in the zonal mean and pre-industrial values are not shown. The latitudinal distribution of the available terrestrial proxies is rather limited and covers mostly middle latitude regions.

A good agreement is seen with proxy data from the North Atlantic (Greenland), South America (Chile), Europe, Australia and the Antarctic margin. While most higher latitude (>40°N) proxies agree fairly well with model temperatures over China and North America, there is a large discrepancy at lower latitudes. As discussed before, continental interiors are seemingly too warm in the model especially in summer. Meanwhile, many of the proxy estimates from China likely underesti-

mate summer temperatures (possibly explaining the apparent lack of a latitudinal temperature gradient) as there is no present-day equivalent of vegetation that can withstand such high heat stresses. In addition, proxy temperatures are representative of a specific location while the horizontal resolution of the simulated atmosphere is limited ($\sim 2°$). Therefore, the model will underestimate local effects by having a strongly smoothed topography.

Using the 0.1° reconstruction from Baatsen et al. (2016), model temperatures are corrected for the



difference with model topography assuming a free atmospheric lapse rate of -6.5K/km. Especially North American sites show a large improvement when correcting for model topography, as shown in the inset of Figure 11 (comparing small squares to coloured circles). The discrepancy remains for the lower latitude part of the Chinese sites, where mean annual air temperatures are probably un-

derestimated by proxies as well as overestimated by the model. Comparing 2x PIC model results to late Eocene proxies shows a similar result (Appendix: Table A4 and Figure A2), with again a good agreement in middle latitude regions but larger differences over interior China.

The model results presented here are again compared to those from Goldner et al. (2014) (GH14)

and shown in the Appendix (Figure A4). Similar to the ocean results, all of the considered Eocene simulations have a comparable zonal mean temperature distribution with the 38Ma cases being generally warmer. High summertime temperatures over low-middle latitude continental regions are more pronounced in the 38Ma 4x PIC case, but also present for 45Ma 4x $CO_2$ from GH14. The large seasonality over Antarctica is also enhanced in the 38Ma simulations compared to GH14, with summer

temperatures well over 30°C in the 4x PIC results. Because of these extreme summertime temperatures on Antarctica and cloud-albedo feedbacks in the Arctic, annual mean (zonal mean) temperatures are globally above 5°C in our 38Ma 4x PIC simulation.

Seasonality in the atmosphere is studied in more detail by comparing the months June-August to December-February for the 4x PIC case as shown in Figure 12. The coldest regions in winter are north-east Siberia and central Antarctica, where average temperatures drop below -10°C. Nearby ocean temperatures keep Antarctic coastal regions mild and at or above freezing in winter. The continental sub-tropics show the highest summer temperatures on Earth, with maxima on average

exceeding 50°C in many locations. Such temperatures seem unrealistically high and do not agree with the vegetation imposed in the model. This issue is probably related to fixed vegetation types, leading to trees losing their foliage in summer and creating a rough surface with very efficient drying and subsequent heating. Despite reduced equator-pole differences compared to the pre-industrial reference, middle latitude regions still exhibit a sharp temperature gradient, especially in winter. This

leads to strong zonal flow and orographic lift seen in pressure and precipitation patterns (Figure 12c-d). The Indo-Pacific ITCZ is always double, but most intense in summer and linked to monsoonal troughs over land. Heavy rains from summer monsoons are most prominent across South-East Asia, (Northern and Southern) Africa and South America.

In winter, plumes of precipitation extend poleward and eastward from the western low latitude

ocean basins. These are typical storm tracks caused by baroclinic instability in the middle latitudes, which are associated with sharp mid-tropospheric temperature gradients and high upper level winds (through the thermal wind balance, Figure 12e-f). Persistent jet streaks around 90°E, linked to sta-



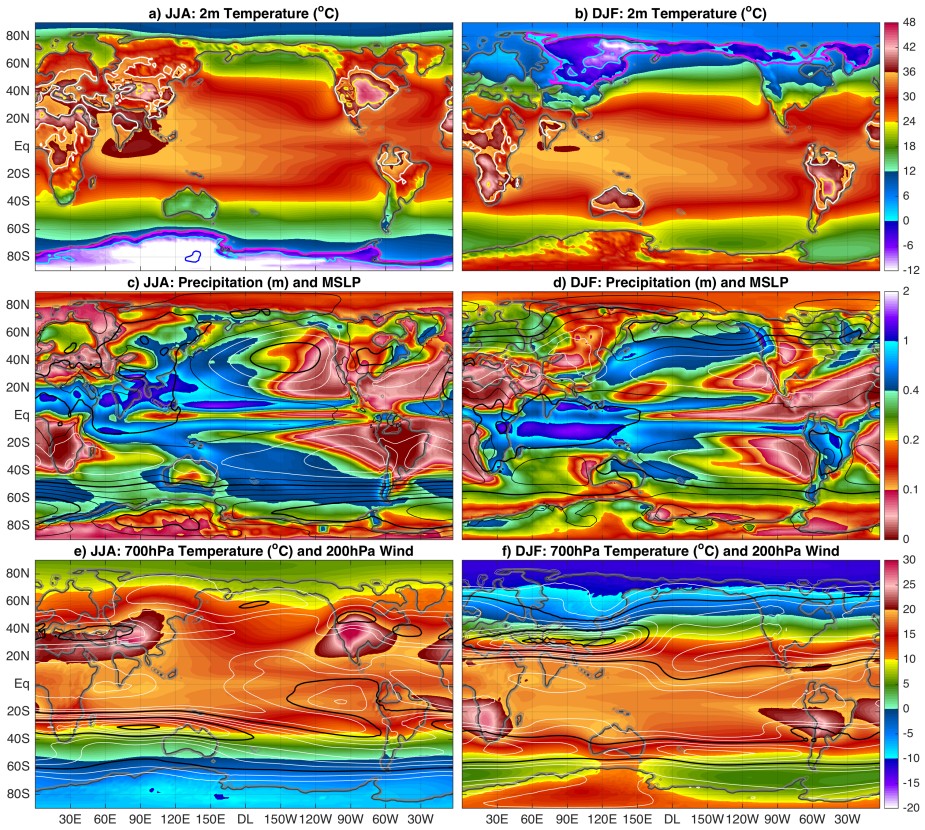

**Figure 12. a)** June-July-August and **b)** December-January-February mean 2m air temperature with contours indicating minimum and maximum values (magenta: $T_{min} < 0°C$, cyan: $T_{min} < -10°C$, blue: $T_{min} < -20°C$, white: $T_{max} > 40°C$ and yellow: $T_{max} > 50°C$), **c)** and **d)** similar for total precipitation and mean sea level pressure, and **e)** and **f)** for 700hPa temperature and 200hPa wind speed. Color scales and contour values are all similar to those used in Figure 9.

tionary Rossby Waves, are most prominent in winter. Smaller stationary waves are seen in summer and linked to sub-tropical ridges where high surface temperatures are mixed upwards to 700hPa. At

this level the effect of solar heating at high altitudes (∼2km) on East-Antarctica stands out, where temperatures are about 10°C higher than those seen over the Southern Ocean.

Finally, some of the peculiar features in the atmospheric circulation of the 4x PIC case are con-

sidered by looking at the zonal mean zonal wind and temperatures for June-August (Figure 13a) and December-February (Figure 13b). The zonal mean temperature shows a typical pattern for the

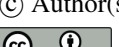



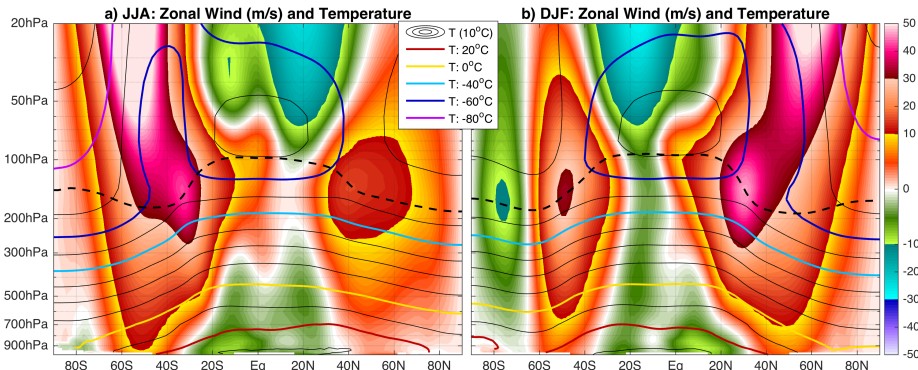

**Figure 13. a)** Zonal average zonal wind (shading) and temperature (contours every $10°$C) for the months June-July-August, **b)** similar to a) for December-January-February. Tropopause height (zonal mean) is indicated by the dashed black line and pressure is used as the vertical coordinate.

Northern Hemisphere summer. In the troposphere, a steady cooling towards the poles can be seen, but in the stratosphere further cooling only takes place on the winter pole because of solar heating at the summer pole. Persistent cold temperatures are present around the top of the tropical tropopause,
at about 70hPa (compared to 90hPa at present-day). Associated with the meridional temperature gradients in the troposphere are the sub-tropical and polar jets, where the former is more pronounced in the zonal mean and both jets are stronger in the winter hemisphere. Thermal wind balance demands an increase of the zonal wind with height as long as temperatures decrease poleward. As a result, the polar jet extends into the stratosphere in the winter hemisphere, while decreasing again above the
tropopause in summer. The polar regions are thus surrounded by a cyclonic polar vortex in winter that extends from the surface to the top of the stratosphere.

An extension of the subtropical jet towards the equator can be seen, resulting in positive (westerly) zonal winds at the equator and thus superrotation at the tropopause even in the annual mean. This is caused by the momentum transfer of atmospheric Rossby waves (causing the 'V' shape in the trop-
ical tropopause in Figure 9d) and the strong winter cell of the Hadley circulation. As it was shown by Caballero and Huber (2010), superrotation does occur in very warm climate simulations and is therefore no surprise to emerge in the 4x PIC case where equatorial SSTs reach $35°$C. Although weaker, zonal mean equatorial westerly winds are still present in the 2x PIC climate. The highest (annual mean, zonal mean) equatorial SSTs in this case are nearly $33°$C and thus close to the thresh-
old found in Caballero and Huber (2010). In the lower tropical troposphere, perennial easterlies are present with two maxima associated with the double ITCZ and strongest winds at the winter hemisphere (corresponding to the stronger Hadley Cell).

A remarkable component of the atmospheric circulation develops in summer over Antarctica. As suggested by temperatures shown in the Figures 12e-f and 13, the meridional temperature gradient





reverses in summer across southern high latitudes. The Antarctic continent thus acts as a (slightly
elevated) heat island, reversing the meridional gradients in both pressure and temperatures and thus
also zonal winds through the thermal wind balance. While the sub-tropical jet remains unaffected
(driven by the middle latitude gradients), the polar jet becomes anticyclonic in both the troposphere
and stratosphere with a maximum near the tropopause. This makes the Antarctic summer climate

resemble that of the sub-tropics rather than a typical polar summer.

## 3.3   Climate Sensitivity

By comparing the 4x to the 2x PIC equilibrium climate states, a measure for the equilibrium climate

sensitivity (ECS) can be obtained. To exclude possible effects of decadal-scale variability (see Fig-
ure 5), 200 years instead of 50 are considered in the climatologies used here. The radiative forcing
associated with doublings of $CO_2$ and $CH_4$ is estimated from the study of Etminan et al. (2016).
Doubling of only $CO_2$ from the pre-industrial value (280 ppm) corresponds to a radiative forcing of
$\Delta RF^{CO_2} = 3.8$ Wm$^{-2}$. The first doubling of both $CO_2$ and $CH_4$ starting from pre-industrial values

results in a combined radiative forcing of $\Delta RF^{2\times} = 4.23$ Wm$^{-2}$ and the second doubling gives
$\Delta RF^{4\times}_{2\times} = 4.48$ Wm$^{-2}$.

In Figures 14a-b the equilibrium temperature difference between the Eocene 4x PIC and 2x PIC
simulations is shown, normalised by the associated radiative forcing ($\Delta RF^{4\times}_{2\times}$). Note that both sim-
ulations were integrated to equilibrium separately, without ever imposing a strong perturbation of

greenhouse gas doubling. Even in this virtually ice-free world significant polar amplification can be
observed, with most of the warming occurring in high latitude continental regions. As in the present
day climate the polar amplification is in part due to albedo effects from vegetation, snow cover
and (limited) seasonal sea ice. Low latitude regions, on the other hand, warm less as they are more
strongly governed by moist processes and tied to SST's.

Averaging the response (to $\Delta RF^{4\times}_{2\times}$) globally yields an equilibrium climate sensitivity of $S_{EO} =$
0.69°C/Wm$^{-2}$ for these simulations. The Eocene ECS is smaller than the reported 0.82°C/Wm$^{-2}$
for the present-day version of the CESM model, using $\Delta RF^{CO_2}$ and $\Delta T = 3.13$°C from Bitz et al.
(2012). The differences in ECS suggests that the net effect of the fast feedback processes is less
strong in the Eocene climate state than in the present day climate (von der Heydt et al., 2014). Note,

however, that the present day ECS is determined from an equilibrium simulation using a slab ocean
model. For the 38Ma Eocene, our simulations indicate a temperature response to a doubling of $CO_2$
and $CH_4$ ranging from a 2°C warming in low latitude regions to as much as 8°C at high latitudes
(∼0.4-1.8 °C/Wm$^{-2}$ in Figure 14a). The zonal variation in the warming signal is highest in the
middle latitude regions because of land-ocean contrasts, while differences in seasonal response are

largest in polar regions.





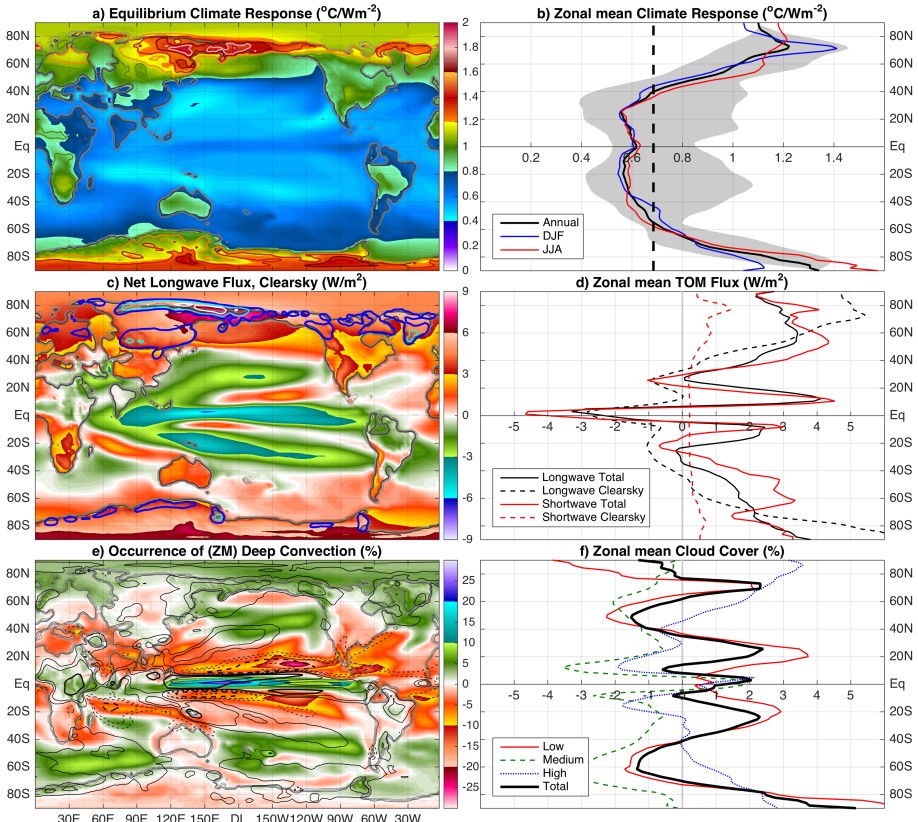

**Figure 14. a)** Annual mean temperature response, normalised per W/m² of the 4x PIC compared to the 2x PIC equilibrium climate with contours showing the winter season only (using the same color scale). **b)** Zonal mean normalised temperature response; annual (black), December-January-February (blue) and June-July-August (red). The grey shading indicates minimum and maximum values for each latitude for the annual mean values. The (area weighted) global average (i.e. climate sensitivity) is $0.69°C/Wm^{-2}$ and is indicated by the black dashed line. **c)** Clear-sky component of the net longwave flux change at the top of model (TOM) for 4x PIC vs. 2x PIC, similar for the shortwave flux in contours (blue: 1W/m², cyan: 2W/m² and white: 5W/m²). **d)** Zonal mean TOM flux change for longwave (black) and shortwave (red) fluxes, corresponding clear-sky components are shown using dashed lines. Note that longwave fluxes are define positive upward while shortwave fluxes are positive downward. **e)** Change in the occurrence of ZM parameterised deep moist convection and difference in precipitation (contours at 100, 200, 500 and 1000 mm, dashed for negative changes and thick line indicating 500mm). **f)** Zonal mean changes in cloud cover (in %) for low (red), medium (green) and high (blue) clouds, the thick black line indicates changes in total coverage.

The polar amplification signal within the middle-late Eocene climate can be explained by looking at



the changes in net fluxes (longwave and shortwave) at the top of model (TOM) as shown in Figures
14c-d. Looking only at the clear-sky component, a large increase in longwave radiation occurs at
high latitudes in correspondence to increased surface temperatures (Planck feedback) in the 4x PIC
case compared to the 2x PIC one. A slight shortwave response can also be seen (contours in Figure
14c) due to local decreases in sea ice cover (Arctic Ocean and Ross Sea) and surface albedo changes
(snow cover and growth season length). A general decrease in longwave flux at low latitudes can be
explained by tropopause responses and stratospheric cooling, resulting in reduced outgoing radia-
tion.

In addition, strong alternating signals of differences in shortwave and longwave radiation occur with
a pronounced minimum at the equator and maxima over the ITCZ locations. This suggests different
patterns of deep convection and thus also precipitation, which are both shown in Figures 14e-f. In-
deed, an increase of convection and precipitation is seen over the equatorial West Pacific, causing
increased cloud cover and therefore a decrease in radiative fluxes (i.e. less outgoing longwave ra-
diation and more reflected shortwave radiation, thus less incoming). In contrast, a decrease of deep
convection and precipitation in the Pacific and South Indian Ocean ITCZ leads to a decrease in mid-
and upper level clouds and thus enhanced shortwave radiation.

Stronger warming in the Pacific cold upwelling tongue (Figure 14a) suggests a shift towards more
El Niño-like conditions in the warmer 38Ma climate. The equatorial westerlies seen in Figure 13
may play an important role in the regional climate through a feature resembling the Madden-Julian
Oscillation, similar to the one found in Caballero and Huber (2010). A general increase in deep con-
vection and precipitation can be seen in most sub-tropical monsoon regions and the middle-latitudes
in the 38Ma Eocene climate. A prominent decrease of low level clouds in the Arctic is also seen
due to the Arctic Ocean warming up and decreasing the strength of a summertime marine inversion.
Over Antarctica, on the other hand, a large increase in low-level cloudiness is related to increased
moisture helping to keep up winter temperatures at the surface (Figures 14e-f).


The Earth System Sensitivity (ESS) for this model set-up can be estimated by comparing the
38Ma Eocene simulations to the pre-industrial reference. In the middle-late Eocene climate not only
greenhouse gases are different, but also the land-sea distribution, orography, vegetation and land ice
cover (Lunt et al., 2010a). The global mean air temperature warming from the pre-industrial ref-
erence to the Eocene 2x PIC simulation together with the radiative forcing due to the first $CO_2$
and $CH_4$ doubling ($\Delta RF^{2\times} = 4.23$ Wm$^{-2}$) results in ESS$^{2\times}$ = 2.36°C/Wm$^{-2}$. Comparing the
Eocene 4x PIC simulation to the pre-industrial reference, on the other hand, gives a lower ESS$^{4\times}$
= 1.49°C/Wm$^{-2}$. Because the radiative forcing of both $CO_2$ and $CH_4$ depends nonlinearly on the
greenhouse gas concentrations, the radiative forcing due to a quadrupling is not equal to the sum





| Simulation Measure | Pre-industrial | 2x PIC 38Ma | 4x PIC 38Ma |
|---|---|---|---|
| $MAT_{glob}$ (°C) | 13.71 | 23.66 | 26.75 |
| $SST_{glob}$ (°C) | 18.35 | 25.83 | 28.36 |
| $SST_{eq}$ (°C) | 26.88 | 31.71 | 33.91 |
| $T_{deep}$ (°C) | 0.7 | 8.6 | 11.5 |

**Table 3.** Average equilibrium temperatures at the end of each simulation, showing $MAT_{glob}$: global mean air temperature (at 2m reference height), $SST_{glob}$: global mean sea surface temperature, $SST_{eq}$: equatorial ($< 23°$N/S) average SST, and $T_{deep}$: global mean deep ocean ocean temperature (below 2000m). These values are calculated using the last 200 years of each simulation.

of two doublings (Etminan et al., 2016). As we only compare model equilibrium states that do not
result from transient quadrupling of greenhouse gases, a linear combination is assumed here such
that: $\Delta RF^{4\times} = \Delta RF^{2\times} + \Delta RF_{2\times}^{4\times} = 8.71 W/m^2$. Using the radiative forcing for a $CO_2$ doubling
($\Delta RF^{CO_2} = 3.8 W/m^2$), the values of ESS determined here correspond to a $5.68 - 8.97 W/m^2$ per
$CO_2$ doubling. The lower of these values (considering the 4x PIC climate) lies close to the 6°C per
$CO_2$ doubling ESS estimate from geological data (Royer et al., 2012).

When comparing global mean temperature increase between the pre-industrial reference and both
Eocene simulations, clearly not all of the warming is a result higher greenhouse gas concentrations.
To separate the effect of palaeogeography (including albedo and vegetation changes, i.e. also land-ice
distribution) from the greenhouse gas driven warming we assume that the 38Ma Eocene equilibrium
climate sensitivity $S_{EO}$ can be calculated using the combined radiative forcing due to greenhouse
gas changes and palaeogeography changes:

$$S_{EO} = \frac{\Delta T}{\Delta RF + G}, \tag{1}$$

where $\Delta T$ is the temperature difference between two climate states, $\Delta R$ the radiative forcing change
from greenhouse gases and $G$ the radiative forcing due to the (integral) effect of palaeogeography.
Note that the latter also includes changes in the distribution of e.g. land ice and vegetation. In order
to be compatible with estimates of ECS and ESS given above, this formulation gives $S$ in normalised
units of [°C/Wm$^{-2}$], instead of [°C per $CO_2$ doubling] as in Royer et al. (2012).

Since both our 38Ma Eocene simulations use the same (except $CO_2$ and $CH_4$) boundary conditions,
it is reasonable to assume that $S$ and $G$ estimated from the comparison of each Eocene simulation
with the pre-industrial case should be similar for both simulations. By comparing both Eocene runs
to the pre-industrial reference, we can then estimate $G$ from:

$$G = \frac{\Delta T^{2\times} \cdot \Delta RF^{4\times} - \Delta T^{4\times} \cdot \Delta RF^{2\times}}{\Delta T^{4\times} - \Delta T^{2\times}}, \tag{2}$$



where $\Delta T^{4\times}$ and $\Delta T^{2\times}$ denote the temperature difference with respect to the pre-industrial climate for the 4x PIC and 2x PIC case, respectively. This leads to an estimate of $G = 10.32 \mathrm{W/m}^2$, and using equation 1 with either $(\Delta T^{4\times}, RF^{4\times})$ or $(\Delta T^{2\times}, RF^{2\times})$ leads to $S = 0.69°\mathrm{C/Wm}^{-2}$. As expected, the value of $\mathrm{S}_{EO}$ is the same as the one found previously when comparing both Eocene simulations to each other.


Alternatively, $G$ can be estimated by comparing a pre-industrial climate under 2x PIC forcing, with the results from the modelled 2x PIC Eocene case. Using the pre-industrial ECS determined for the model ($\mathrm{S}_{PI} = 0.82°\mathrm{C/Wm}^{-2}$) and radiative forcing of the first doubling of greenhouse gases ($\Delta RF^{2\times}$, a temperature change of 3.48°C would be expected. As the change in global mean tem-

perature between the pre-industrial reference and 2x PIC Eocene climate is 9.97°C, this leaves an additional warming of 6.47°C owing to integrated global geography changes. In order to convert this warming into an estimate of $G$, we can use either the Eocene $\mathrm{S}_{EO}$ or pre-industrial $\mathrm{S}_{PI}$ giving a possible range of $G = 7.86 - 9.30 \ W/m^2$.

Finally, we can only consider equatorial ($< 23°$N/S) SST's ($\mathrm{SST}_{eq}$; with a 3/2 ratio between global and equatorial temperature change to account for polar amplification, as discussed in Royer et al. (2012)) or deep sea temperatures ($\mathrm{T}_{deep}$) to be more compatible with ECS estimates from proxy data. An overview of used temperature differences: $\mathrm{SST}_{eq}$ and $\mathrm{T}_{deep}$ in addition $\mathrm{MAT}_{glob}$ to (global mean, annual mean 2m temperature, used above) and the resulting values for $\mathrm{S}_{EO}$ and $G$ is given

in Table 4. Not surprisingly, the estimate of $G$ decreases when using only ocean temperatures as the direct effect of cooler temperatures over land ice is removed. The results for $\mathrm{S}_{EO}$ show less variation between different methods and are tied to the respective temperature changes between the 2x PIC and 4x PIC cases.

While considering global MAT changes will likely overestimate the effect of global geography

changes, the opposite is true for equatorial SST's. Still we find a possible range of $G = 5.61 - 10.23 \ W/m^2$, equivalent to ~1.5-2.7 $CO_2$ doublings with a most likely value around 2 doublings or a ~6°C warming. This is considerably higher than the estimated 1.8°C warming related to geography (using HadCM) found by Lunt et al. (2012) and more than the ~2-4W/m$^2$ (i.e. 0.6-1.1 $CO_2$ doublings) suggested by Royer et al. (2012). A considerably higher warming of ~5°C from integral

geography effects in the Eocene has been found by Caballero and Huber (2013) that matches up with our estimates. Finally, higher estimates of 2-6°C can also be deduced using the different models used in Lunt et al. (2012).



| Method | $\Delta T^{4\times}$ (°C) | $\Delta T^{2\times}$ (°C) | $S_{EO}$ (°C/Wm$^{-2}$) | $G$ (W/m$^2$) |
|---|---|---|---|---|
| $\mathbf{MAT}_{glob}$ | 13.0 | 9.97 | 0.686 | 10.32 |
| $\mathbf{SST}_{eq}$ | 10.5 | 7.22 | 0.736 | 5.61 |
| $\mathbf{T}_{deep}$ | 10.8 | 7.9 | 0.647 | 7.99 |

**Table 4.** Temperature differences comparing the 4x PIC ($\Delta T^{4\times}$) and 2x PIC ($\Delta T^{2\times}$) Eocene climate to the pre-industrial reference, derived values for equilibrium climate sensitivity ($S$) and forcing from (integral) geography changes ($G$) using equation 2. Results are shown using global mean 2m temperature, equatorial SST's (with a 3/2 ratio, as discussed in Royer et al. (2012)) and deep sea temperature (below 2000m).

## 4 Summary and Conclusions

Using version 1.0.5 of the Community Earth System Model (CESM), we presented results of simulated 38Ma Eocene climates at both high (4x PIC) and low (2x PIC) concentrations of $CO_2$ and $CH_4$, using the most recent palaeogeographic constraints. These are among the first simulations with a fully-coupled and detailed climate model to study the middle-late Eocene climate, using a new 38Ma geography reconstruction at high resolution.

The 4x PIC case shows a warm climate with a global mean surface temperature of about 27°C (pre-industrial: ∼14°C) and a low equator to pole temperature gradient. The global heat budget is approximately meridionally symmetric, which is reflected by the zonal mean temperature pattern. Deep water formation occurs in the South Pacific Ocean, while the North Atlantic is stably stratified and stagnant due to the outflow of brackish Arctic waters. A shallow and rather weak precursor of an Antarctic Circumpolar Current is present, mainly driven by seasonally dependent wind stresses. Continental low-middle latitude regions are characterised by high seasonality on both hemispheres and strong summer monsoons. Middle and high latitudes mostly have mild winters, warm summers and pronounced storm tracks. The Arctic is rather cool due to its geographic isolation and the Antarctic continent shows strong seasonality with especially hot summers.

Comparing the 4x PIC simulation to the available 42-38 Ma sea surface and terrestrial temperature proxy records, the simulated climate makes a fairly good match with middle-to-high latitudes without the tropics being too hot. This indicates that the CESM is able to simulate a warm greenhouse climate of the late-middle Eocene (Bartonian) without the need for extremely high (>1200ppm $CO_2$) atmospheric carbon. In the 2x PIC simulation, the patterns of oceanic and atmospheric equilibrium circulation are qualitatively very similar to those of the 4x PIC one. Based on a similar comparison between model results and 38-34 Ma proxy temperature estimates, the 2x PIC case presented here is a good analog for the late Eocene climate (Priabonian).



Previous Eocene simulations (at 4x PI $CO_2$) with the same model but a different (45Ma) continental geography and lower resolution have resulted in overall similar sea surface temperature distributions (Goldner et al., 2014). However, the 38Ma 4x PIC case presented here is about 4-5°C warmer

globally in both SST's and land temperatures. Higher resolution and a more specific geography reconstruction allows for a better representation of regional features, including equatorial upwelling, zonal heterogeneity in the Southern Ocean and Antarctic summer warmth. The increase in temperature is mainly a result of a quadrupling of methane (resulting in a 4.7x pre-industrial $CO_2$ equivalent radiative forcing), different atmospheric grids (finite volume versus spectral) and higher spatial res-

olution in combination with a different geography.

Comparing the 38Ma 2x and 4x PIC cases, an equilibrium climate sensitivity $S_{EO} = 0.69°C/Wm^{-2}$ was found, which is slightly lower than the same model's present-day value. The difference in equilibrium climate sensitivity between the Eocene and the present day reflects a state-dependence of

the fast feedback processes allowing the Eocene climate to respond differently to a greenhouse gas doubling than the present day climate.

When also taking the pre-industrial reference simulation into consideration, we find a fixed forcing $G = 8.65W/m^2$ from (integral) geography changes. Previous studies have noted this effect in

terms of an offset in global mean temperature between pre-industrial and palaeo simulations (Caballero and Huber, 2013; Lunt et al., 2012). Similar to what has been found in Caballero and Huber (2013) the direct effect of ice sheet distribution is limited, leaving a considerable warming due to other geography related changes. When using oceanic instead of atmospheric temperatures, the influence of topography and land surface changes is reduced mostly by the direct effect of the ice sheets

and vegetation changes. Although smaller, the estimate for $G$ is still larger than suggested in most previous studies. This indicates a major contribution to $G$ from changes in continental geometry and the related circulation patterns, that these simulations are better able to resolve.

Several peculiar (extreme) phenomena were found in the simulations. Extremely high (∼50°C) sum-

mer temperatures occur in the sub-tropics and are probably related to fixed vegetation types. Strong seasonality is also seen on Antarctica, where summer temperatures reach up to 35°C in the 4x PIC case. The absence of an ice sheet, warm waters surrounding the continent and summertime insolation cause te Antarctic continent to become a heat island. Sea ice coverage is very limited and only occurs sporadically during the coldest months for both 38Ma Eocene cases. Even without sea

ice or extensive snow cover, significant polar amplification is seen for a doubling in $CO_2$ and $CH_4$. With limited surface albedo feedbacks, changes in cloud coverage and radiative forcing are the main drivers behind the further reduction of the meridional temperature gradient.





As the simulated middle-late Eocene (4x PIC) climate is in reasonable agreement with estimates
from proxy records, the simulation results may be used to try to interpret proxy records in more
detail.

*Acknowledgements.*  This work was carried out under the program of the Netherlands Earth System Science
Centre (NESSC), financially supported by the Dutch Ministry of Education, Culture and Science. The com-
putations using CESM1.0.5 were done on the Cartesius at SURFsara in Amsterdam. The use of the SURFsara
computing facilities was sponsored by NWO-EW (Netherlands Organisation for Scientific Research, Exact Sci-
ences) under the project 15508. AvdH acknowledges travel support to network partners from the EPSRC-funded
Past Earth Network (Grant number EP/M008363/1).





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




## Appendix A: Sea surface temperature proxies

**42-38 Ma SST proxies:**

| Site | 38Ma lat | 38Ma lon | SST (°C) | $\sigma$ (°C) | Reference | Method |
|---|---|---|---|---|---|---|
| DSDP 277 | -55.5 | 178.1 | 28 | 4.7 | Hines et al. (2017) | Mg/Ca |
| ODP 865 | 11.8 | -147.0 | 31 | 4.7 | Tripati et al. (2003) | Mg/Ca |
| ODP 925 | 3.5 | -30.9 | 32 | 2.5 | Liu et al. (2009) | $TEX_{86}^H$ |
| ODP 929 | -1.8 | -31.2 | 30 | 2.5 | Liu et al. (2009) | $TEX_{86}^H$ |
| ODP 1052 | 24.2 | -61.6 | 32 | 1.2 | Okafor et al. (2009) | Mg/Ca |
| ODP 1172 | -57.8 | 157.9 | 24 | 2.5 | Bijl et al. (2009) | $TEX_{86}^H$ |
| ACEX | 87.9 | 136.2 | 82.5 | -5.7 | Evans et al. (2018) | $UK_{37}$ |
| Gulf Coast, US | 28.3 | -74.1 | 26 | 1.4 | Kobashi et al. (2004) | $\delta^{18}O$ |
| Istra More 5 | 34.1 | 13.1 | 24 | 0.7 | Pearson et al. (2001) | $\delta^{18}O$ |
| Java KW01 | 0.1 | 114.5 | 35 | 2.0 | Evans et al. (2018) | $\Delta_{47}$ |
| | | | 36.3 | 1.9 | Evans et al. (2018) | $\Delta_{47}$ |
| New Zealand | -49.3 | -169.7 | 26 | 4.7 | Hines et al. (2017) | Mg/Ca |
| Seymour Island | -68.5 | -62.5 | 12.7 | 2.4 | Douglas et al. (2014) | $\Delta_{47}$ |
| | | | 13.1 | 2.4 | Douglas et al. (2014) | $\Delta_{47}$ |
| | | | 20.3 | 2.5 | Douglas et al. (2014) | $TEX_{86}^H$ |
| Tanzania | -16.6 | 40.6 | 34 | 2.5 | Pearson et al. (2007) | $TEX_{86}^H$ |
| Tanzania Lindi | -17.6 | 40.6 | 32 | 0.7 | Pearson et al. (2001) | $\delta^{18}O$ |
| Hampshire Basin, UK | 43.6 | -2.9 | 23.2 | 2.6 | Evans et al. (2018) | $\Delta_{47}$ |

**Table A1.** Overview of 42-38 Ma SST proxies; site location, 38Ma reconstructed coordinates, SST estimate, calibration error, reference and method used.



**38-34 Ma SST proxies:**

| Site | 38Ma lat | 38Ma lon | SST (°C) | $\sigma$ (°C) | Reference | Method |
|---|---|---|---|---|---|---|
| DSDP 277 | -55.5 | 178.1 | 26.6 | 2.5 | Liu et al. (2009) | $TEX_{86}^H$ |
| | -55.5 | 178.1 | 25.6 | 4.7 | Liu et al. (2009) | $UK_{37}$ |
| DSDP 336 | 55.7 | -11.0 | 20 | 1.5 | Liu et al. (2009) | $UK_{37}$ |
| DSDP 511 | -58.8 | -30.4 | 19.5 | 2.5 | Liu et al. (2009) | $TEX_{86}^H$ |
| | | | 19.6 | 1.5 | Liu et al. (2009) | $UK_{37}$ |
| ODP 689 | -68.3 | 14.0 | 12.3 | 4.6 | Petersen and Schrag (2015) | $\Delta_{47}$ |
| | | | 22.0 | 1.5 | Petersen and Schrag (2015) | $\Delta_{47}$ |
| ODP 803 | -0.8 | -169.1 | 27.4 | 2.5 | Liu et al. (2009) | $TEX_{86}^H$ |
| ODP 913 | 66.2 | -2.4 | 18.3 | 1.5 | Liu et al. (2009) | $UK_{37}$ |
| ODP 925 | 3.5 | -30.9 | 30 | 2.5 | Liu et al. (2009) | $TEX_{86}^H$ |
| ODP 929 | -1.8 | -31.2 | 29 | 2.5 | Liu et al. (2009) | $TEX_{86}^H$ |
| ODP 998 | 14.6 | -69.4 | 22.1 | 2.5 | Liu et al. (2009) | $TEX_{86}^H$ |
| ODP 1052 | 24.2 | -61.6 | 30 | 1.2 | Okafor et al. (2009) | Mg/Ca |
| ODP 1090 | -51.9 | 157.9 | 22 | 2.5 | Bijl et al. (2009) | $UK_{37}$ |
| ODP 1172 | -57.8 | 157.9 | 22 | 2.5 | Bijl et al. (2009) | $TEX_{86}^H$ |
| Alabama, US | 27.6 | -72.4 | 27 | 0.7 | Pearson et al. (2001) | $\delta^{18}O$ |
| Browns Creek, Aus | -53.8 | 147.2 | 18.9 | 0.7 | Kamp et al. (1990) | $\delta^{18}O$ |
| Gulf Coast, US | 28.3 | -74.1 | 22 | 1.4 | Kobashi et al. (2004) | $\delta^{18}O$ |
| New Zealand | -49.3 | -169.7 | 30.9 | 4.7 | Hines et al. (2017) | Mg/Ca |
| Panama | 0.9 | -72.9 | 32 | 1.4 | Tripati and Zachos (2002) | $\delta^{18}O$ |
| Seymour Island | -68.5 | -62.5 | 12.2 | 2.4 | Douglas et al. (2014) | $\Delta_{47}$ |
| | | | 13.0 | 3.0 | Douglas et al. (2014) | $\Delta_{47}$ |
| | | | 16.0 | 2.5 | Douglas et al. (2014) | $TEX_{86}^H$ |
| St Stephens Quarry | 27.2 | -72.2 | 32.5 | 2.2 | Wade et al. (2012) | Mg/Ca |
| | | | 34.9 | 4.7 | Wade et al. (2012) | Mg/Ca |
| | | | 30.5 | 2.5 | Wade et al. (2012) | $TEX_{86}^H$ |
| Tanzania | -16.6 | 40.6 | 29.5 | 0.7 | Pearson et al. (2007) | $\delta^{18}O$ |
| | | | 33.0 | 2.5 | Pearson et al. (2007) | $TEX_{86}^H$ |
| | | | 29.7 | 3.2 | Evans et al. (2018) | $\Delta_{47}$ |
| Tanzania Lindi | -17.6 | 40.6 | 30.0 | 0.7 | Pearson et al. (2001) | $\delta^{18}O$ |

**Table A2.** Overview of 38-34 Ma SST proxies; site location, 38Ma reconstructed coordinates, SST estimate, calibration error, reference and method used.





**Appendix B:  Terrestrial temperature proxies**

**Late-middle Eocene land temperature proxies:**

| Site | $T_{avg}$ (°C) | $T_{max}$ (°C) | $T_{min}$ (°C) | 38Ma lat | 38Ma lon | Reference |
|---|---|---|---|---|---|---|
| **North Atlantic:** | | | | | | |
| ODP 913 MBT | 14.5 | 16.0 | 12.5 | 66.2 | -2.4 | Schouten et al. (2008) |
| ODP 913 Pollen | 13.3 | 16.2 | 11.8 | 66.2 | -2.4 | Eldrett et al. (2009) |
| **North America:** | | | | | | |
| Florissant CO | 14.6 | 17.5 | 10.7 | 37.0 | -86.9 | Boyle et al. (2008) |
| Comstock (OR) | 21 | 22.4 | 19.6 | 44.7 | -102.9 | Retallack et al. (2004) |
| Copper Basin (NV) | 9.5 | 10.5 | 8.5 | 41.5 | -96.0 | Wolfe et al. (1998) |
| Badger's Nose (CA) | 14.5 | 17.1 | 12.5 | 42.1 | -100.6 | Prothero (2008) |
| Sevier (UT) | 13.2 | 13.2 | 13.2 | 38.7 | -94.7 | Gregory-Wodzicki (1997) |
| Gray Butte (OR) | 14.0 | 19.3 | 8.8 | 45.0 | -100.6 | Smith et al. (1998) |
| **South America:** | | | | | | |
| Ñirihuau (Chile) | 18.4 | 18.4 | 18.4 | -49.4 | -59.5 | Hinojosa and Villagrán (2005) |
| **Europe:** | | | | | | |
| Stare Sedlo | 23.9 | 29 | 21.3 | 43.71 | 9.3 | Uhl et al. (2007) |
| Weiße Elster | 22.6 | 24 | 18.7 | 44.9 | 8.3 | Uhl et al. (2007) |
| **China:** | | | | | | |
| Dalianhe | 16.5 | 16.5 | 16.5 | 54.2 | 129.1 | Quan et al. (2012) |
| Huanghua | 18.1 | 18.3 | 17.9 | 52.7 | 129.4 | Quan et al. (2012) |
| Jushu | 15.9 | 16.1 | 15.6 | 52.4 | 126.5 | Quan et al. (2012) |
| Hunchun | 17.5 | 18.4 | 16.5 | 50.8 | 130.9 | Quan et al. (2012) |
| Huadian | 17.0 | 18.4 | 15.6 | 50.8 | 126.4 | Quan et al. (2012) |
| Jijuntun | 16.8 | 17 | 16.5 | 49.6 | 123.3 | Quan et al. (2012) |
| Xilutan | 17.0 | 18.4 | 15.6 | 49.6 | 123.3 | Quan et al. (2012) |
| Kongdian | 17.5 | 18.4 | 16.5 | 47.4 | 118.7 | Quan et al. (2012) |
| Shahejie | 17.1 | 18.4 | 15.7 | 45.6 | 116.3 | Quan et al. (2012) |
| Sanduo | 17.1 | 18.4 | 15.7 | 40.7 | 118.4 | Quan et al. (2012) |
| Dingyuan | 15.5 | 16.1 | 14.8 | 39.6 | 116.1 | Quan et al. (2012) |
| Shuangta | 16.1 | 16.4 | 15.7 | 38.8 | 117.9 | Quan et al. (2012) |
| Qingjiang | 16.5 | 16.5 | 16.5 | 35.5 | 115.5 | Quan et al. (2012) |
| Wenzhou | 17.6 | 18.4 | 16.8 | 34.6 | 121.8 | Quan et al. (2012) |
| Tantou | 17.1 | 18.4 | 15.7 | 40.7 | 110.7 | Quan et al. (2012) |

**Table A3.** Overview of late-middle Eocene terrestrial temperature proxies; site location, mean annual temperature, highest annual temperature, lowest annual temperature and 38Ma reconstructed coordinates.





| Site | $T_{avg}$ (°C) | $T_{max}$ (°C) | $T_{min}$ (°C) | 38Ma lat | 38Ma lon | Reference |
|---|---|---|---|---|---|---|
| Unnamed Unit 1 | 15.9 | 18.4 | 13.3 | 45.6 | 105.8 | Quan et al. (2012) |
| Unnamed Unit 2 | 13.7 | 16.1 | 11.3 | 42.5 | 101.8 | Quan et al. (2012) |
| Honggou | 17.2 | 21.1 | 13.3 | 42.6 | 99.9 | Quan et al. (2012) |
| Relu | 14.1 | 16.4 | 11.7 | 36.3 | 95.1 | Quan et al. (2012) |
| Huoshaogou | 16.4 | 21.7 | 11.0 | 45.4 | 93.7 | Quan et al. (2012) |
| Lulehe | 17.1 | 18.4 | 15.7 | 41.2 | 86.0 | Quan et al. (2012) |
| Lulehe | 17.1 | 18.4 | 15.7 | 37.0 | 88.1 | Quan et al. (2012) |
| Totohe | 14.9 | 16.5 | 13.3 | 34.6 | 87.3 | Quan et al. (2012) |
| Wulagen | 18.2 | 20.8 | 15.6 | 39.3 | 74.4 | Quan et al. (2012) |
| Shenhu | 16.6 | 21.7 | 11.5 | 30.0 | 113.3 | Quan et al. (2012) |
| Liushagang | 17.6 | 23.9 | 11.3 | 28.1 | 109.7 | Quan et al. (2012) |
| Changchang | 18.9 | 20.8 | 17 | 26.8 | 110.6 | Quan et al. (2012) |
| **Australia:** | | | | | | |
| Anglesea | 16.2 | 19.3 | 13.9 | -53.4 | 148.2 | Greenwood et al. (2004) |
| Hasties | 16.2 | 16.2 | 16.2 | -55.3 | 153.8 | Greenwood and Wing (1995) |
| West Dale | 16.1 | 17.9 | 14.2 | -49.1 | 111.4 | Greenwood et al. (2004) |
| **Antarctica:** | | | | | | |
| McMurdo | 13.0 | 13.0 | 13.0 | -72.1 | 158.2 | Passchier et al. (2013) |
| SA Islands AP | 12.0 | 15.0 | 10.5 | -67.5 | -63.5 | Passchier et al. (2013) |

**Late Eocene land temperature proxies:**

| Site | $T_{avg}$ (°C) | $T_{max}$ (°C) | $T_{min}$ (°C) | 38Ma lat | 38Ma lon | Reference |
|---|---|---|---|---|---|---|
| **North Atlantic:** | | | | | | |
| ODP 913 MBT | 14.0 | 15.9 | 7.9 | 66.2 | -2.4 | Schouten et al. (2008) |
| ODP 913 Pollen | 13.4 | 16.2 | 11.8 | 66.2 | -2.4 | Eldrett et al. (2009) |
| **North America:** | | | | | | |
| Florissant CO | 14.6 | 17.5 | 10.7 | 37.0 | -86.9 | Boyle et al. (2008) |
| Comstock (OR) | 21 | 22.4 | 19.6 | 44.7 | -102.9 | Retallack et al. (2004) |
| Copper Basin (NV) | 9.5 | 10.5 | 8.5 | 41.5 | -96.0 | Wolfe et al. (1998) |
| Badger's Nose (CA) | 14.5 | 17.1 | 12.5 | 42.1 | -100.6 | Prothero (2008) |
| Sevier (UT) | 13.2 | 13.2 | 13.2 | 38.7 | -94.7 | Gregory-Wodzicki (1997) |
| Gray Butte (OR) | 14.0 | 19.3 | 8.8 | 45.0 | -100.6 | Smith et al. (1998) |

**Table A4.** Overview of late Eocene terrestrial temperature proxies; site location, mean annual temperature, highest annual temperature, lowest annual temperature and 38Ma reconstructed coordinates.





| Site | $T_{avg}$ (°C) | $T_{max}$ (°C) | $T_{min}$ (°C) | 38Ma lat | 38Ma lon | Reference |
|---|---|---|---|---|---|---|
| **South America:** | | | | -49.4 | -59.5 | |
| Ñirihuau (Chile) | 18.4 | 18.4 | 18.4 | -49.4 | -59.5 | Hinojosa and Villagrán (2005) |
| **Europe:** | | | | | | |
| Stare Sedlo | 23.9 | 29 | 21.3 | 43.71 | 9.3 | Uhl et al. (2007) |
| Weiße Elster | 22.6 | 24 | 18.7 | 44.9 | 8.3 | Uhl et al. (2007) |
| **China:** | | | | | | |
| Hunchun | 18.2 | 18.4 | 17.9 | 50.8 | 130.9 | Quan et al. (2012) |
| Gengjiajie | 18.2 | 18.4 | 17.9 | 49.6 | 123.3 | Quan et al. (2012) |
| Shahejie | 17.5 | 18.4 | 16.5 | 47.4 | 118.7 | Quan et al. (2012) |
| Sanduo | 18.2 | 20.8 | 15.6 | 40.7 | 118.4 | Quan et al. (2012) |
| Dingyuan | 13.9 | 16.1 | 11.6 | 39.6 | 116.1 | Quan et al. (2012) |
| Linjiang | 18.0 | 19.4 | 16.5 | 35.5 | 115.5 | Quan et al. (2012) |
| Pinghu | 17.0 | 18.4 | 15.6 | 34.6 | 121.8 | Quan et al. (2012) |
| Unnamed Unit 2 | 14.1 | 16.5 | 11.6 | 42.5 | 101.8 | Quan et al. (2012) |
| Honggou | 17.2 | 21.1 | 13.3 | 42.6 | 99.9 | Quan et al. (2012) |
| Relu | 15.2 | 15.6 | 14.8 | 36.3 | 95.1 | Quan et al. (2012) |
| Huoshaogou | 15.6 | 21.7 | 9.4 | 45.4 | 93.7 | Quan et al. (2012) |
| Xiaganchaigou | 17.1 | 18.4 | 15.7 | 41.2 | 86.0 | Quan et al. (2012) |
| Wanbaogou | 19.2 | 21.9 | 16.5 | 37.0 | 88.1 | Quan et al. (2012) |
| Bashibulake | 17.1 | 20.8 | 13.3 | 39.3 | 74.4 | Quan et al. (2012) |
| Xiaokuzibai | 18.8 | 20.8 | 16.8 | 43.8 | 78.7 | Quan et al. (2012) |
| Youganwo | 17.2 | 18.6 | 15.7 | 28.6 | 110.8 | Quan et al. (2012) |
| Liushagang | 16.1 | 20.8 | 11.3 | 28.1 | 109.7 | Quan et al. (2012) |
| Yongning Gr. | 17.8 | 18.4 | 17.2 | 28.8 | 106.8 | Quan et al. (2012) |
| Nadu | 15.9 | 16.1 | 15.7 | 30.5 | 106.0 | Quan et al. (2012) |
| Dagzhuka | 17.5 | 18.4 | 16.5 | 29.9 | 87.6 | Quan et al. (2012) |
| **Antarctica:** | | | | | | |
| McMurdo | 13.0 | 13.0 | 13.0 | -72.1 | 158.2 | Passchier et al. (2013) |
| King George | 13.4 | 15.0 | 11.7 | -66.3 | -64.5 | Passchier et al. (2013) |
| ODP 1166 | 12.0 | 12.0 | 12.0 | -64.8 | 87.3 | Passchier et al. (2013) |





**Appendix C:  2x PIC model - proxy temperature comparison**

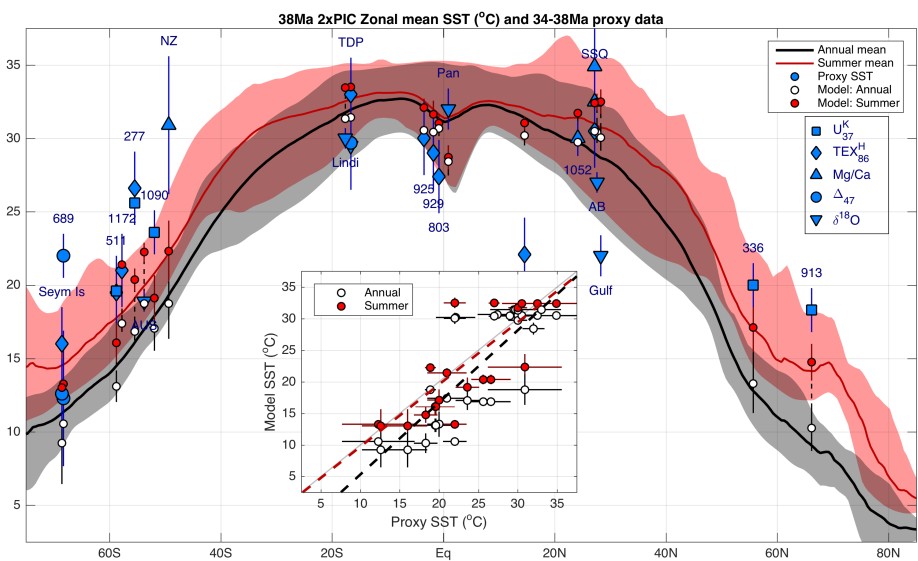

**Figure A1.** Zonal mean, annual mean (black) and summer mean (red) sea surface temperature (SST) with shaded regions showing minimum and maximum values for each latitude (2x PIC simulation). Blue markers indicate estimates from 34-48 Ma SST proxies, whereas the white (annual) and red (summer) circles depict model values at the corresponding 38Ma locations. Error bars are obtained using proxy calibration errors and the spacial variation within a $4° \times 4°$ box surrounding the corresponding location in the model. The inset shows a scatter plot comparing proxy and model SST, with dashed lines indicating a linear fit using annual mean (black) or summer mean (red; only outside of the tropics) model temperatures.



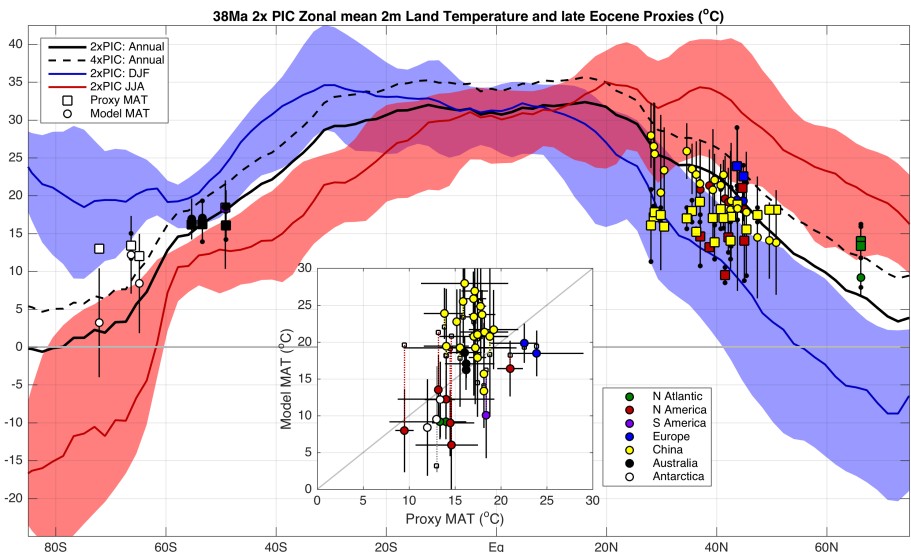

**Figure A2.** Annual mean, zonal mean (land-only) 2m air temperature for the 38Ma 2x PIC equilibrium climate (black line). Zonal mean temperatures, averaged for June-July-August and December-January-February are shown in red and blue, respectively, with their longitudinal range indicated by shaded areas. Markers indicate late Eocene proxy estimates (squares) and corresponding model mean annual temperature (circles). Proxy temperatures are colour coded for their region and error bars (black dots) are indicative for the spread at each site. Similar to model SST, error estimates (black lines) show the spacial variation within a $5° \times 4°$ box surrounding the corresponding location in the model. The inset shows a scatter plot comparing proxy and model air temperatures, where the latter are corrected for differences in model and reconstructed topography (uncorrected: small white squares).





**Appendix D:  4x PIC model - model comparison**

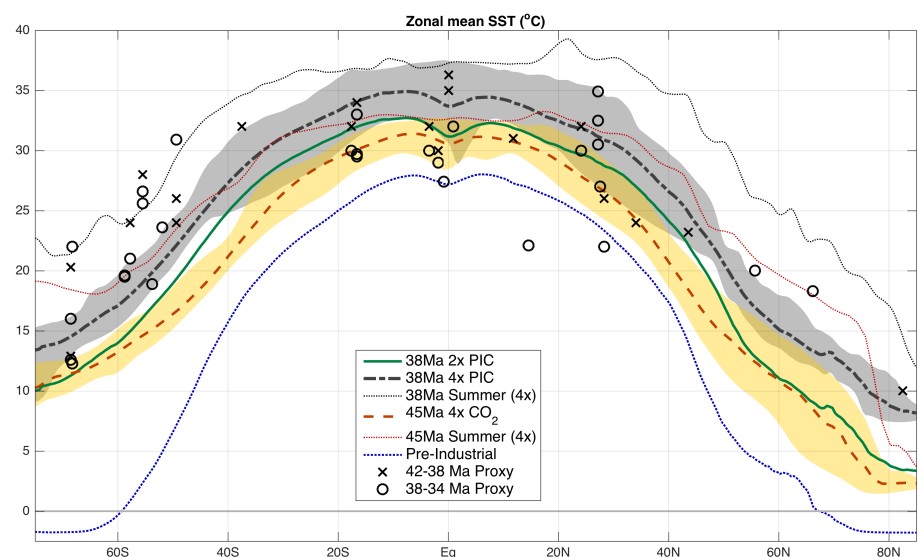

**Figure A3.** Annual mean, zonal mean sea surface temperature for the 38Ma 4x PIC (thick black) and 2x PIC (thick green), 45Ma 4x $CO_2$ (thick red; GH14 Goldner et al. 2014) and pre-industrial reference (blue) simulations. Estimates from proxy data are represented by crosses for 42-38 Ma and circles for the 38-34 Ma period in accordance with other figures. Shading indicates the zonal range in temperatures for both the 38Ma 4x PIC (grey) and 45Ma (yellow) case, while thin dashed lines show the highest summer temperatures at each latitude.





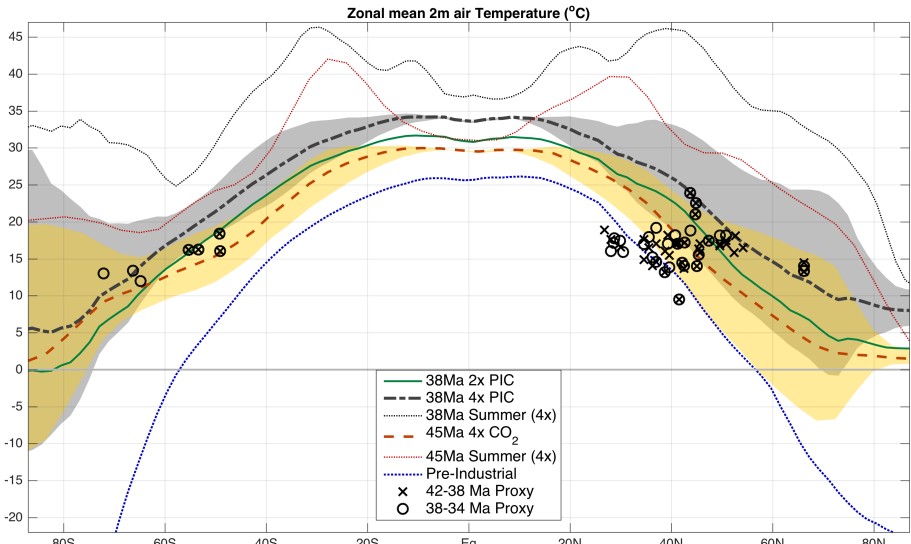

**Figure A4.** Annual mean, zonal mean near-surface (2m) air temperature for the 38Ma 4x PIC (thick black) and 2x PIC(thick green), 45Ma 4x $CO_2$ (thick red; GH14) and pre-industrial reference (blue) simulations. Estimates from proxy data are represented by crosses for 42-38 Ma and circles for the 38-34 Ma period in accordance with other figures. Shading indicates the seasonal (DJF, JJA) range in temperatures for both the 38Ma 4x PIC (grey) and 45Ma yellow) case, while thin dashed lines show the highest summer temperatures at each latitude.