# Peer review of "Equilibrium state and sensitivity of the simulated middle-to-late Eocene climate"

_Climate of the Past, 2018_

## Referee Comment (RC1) · Anonymous Referee #1 · 18 Jul 2018

This manuscript presented simulations of the middle-to-late Eocene climate using the CESM1.0. The model resolution was $\sim$1-2°, higher than most of the previous Eocene simulations. A new set of geographical boundary conditions was employed. The authors claimed that their modeling results were in good agreement with proxy records. They also described many aspects of ocean/atmosphere states and circulations and climate sensitivity in the simulations.

Simulations in the study could potentially contribute to our understanding of the past warm worlds, but I do not find this manuscript in the current form satisfactory. To make this manuscript more accessible to readers, the writing should be improved greatly. Vague expressions should be avoided. To substantiate many of the authors' arguments, new analyses need to be conducted. Please see detailed comments below.

Major comments: 1. The authors used many vague expressions in their manuscript. I only list a few here for illustrative purpose. I suggest the authors carefully go through the manuscript and improve the writing.

Line 285: ". . . in good agreement with the model."

Line 286–287: "A mixed agreement is seen at other latitudes, with model results being too warm in the northern middle latitudes, too cold in southern middle latitudes and good at high latitudes."

Line 300: "Good agreement between proxy and model results . . ."

Line 442: ". . . agree fairly well . . ."

Line 674: ". . . makes a fairly good match . . ."

2. This is a long manuscript with 18 items (figures + tables). I would suggest the authors simplify the figures, shorten the results but highlight and provide in-depth analyses on the differences from previous modeling results. For example, in Figure 1-2, it is not very interesting to show the time series of individual basins (Pacific and Atlantic). In fact, this detailed information is not much discussed in the manuscript. I suggest the authors focus more on the relevant findings and major difference from previous simulations, to make this manuscript more interesting to read.

3. The authors need to do a better job describing their model and experimental setup, to ensure that results are reproducible by others. For example, CESM1.0.5 has active land and sea ice model, which the authors did not mention in the manuscript. Also, what values are used for the orbital parameters, solar constant, and other greenhouse gases (N2O and CFCs)? How is the land surface configured? How are the lakes and rivers treated? What is the soil color in the Eocene simulations? How is the ocean mixing (including tidal dissipation) parameterized? I know this list is long, but it is essential for other researchers to reproduce and correctly interpret results in this study.

4. I find results on model-data comparison and analyses on climate sensitivity more

interesting and relevant than descriptions of many detailed aspects of atmospheric and oceanic state/circulations. I think, when describing the modeling results, after a brief discussion of the model spin-up process, it is better to show modelled results compared with proxy records first. This essentially gives readers a general ideal about the model performance and its advantage/disadvantage compared with previous modeling results.

5. Aside from the analyses on climate sensitivity, this manuscript is highly descriptive with many statements requiring more in-depth analysis to substantiate.

Example 1: when explaining the temperature difference between the present study and Goldner et al. (2014), the authors listed a few possible factors (e.g., model resolution, dynamic core, radiative forcing from CH4, aerosol, ...) but fail to, at least try to, quantify contribution from any factor.

Example 2: the authors ascribed the different climate sensitivity between the Eocene climate and the present day to "fast feedback processes". They failed to explain further what these fast feedback processes are? Are there other processes involved, like clouds, albedo and lapse rate? Have you quantified the feedback strength (e.g., using the partial radiative perturbation) to make the argument?

Example 3: Around Line 22–23, the authors stated that "... without the need for greenhouse gas concentrations much higher than proxy estimates". What are the estimated ranges of CO2 and CH4 from reconstructions? How different are these reconstructions compared with values used in simulations in this study?

There are many other unsubstantiated statements all over the manuscript.

6. Using simplified equations in Etminan et al. (2016) for radiative forcing calculation in a 3D climate model is not justified. For example, the adjusted radiative forcing for a CO2 doubling at preindustrial is 3.5 W m-2 from off-line radiation calculation (Bitz et al., 2012), rather than 3.8 W m-2 from equations in Etminan et al. (2016). It is not

clear how much the difference could be under high CO2 and CH4 levels. The authors need either to calculate the radiative forcing from the off-line version of CESM radiation code, or to conduct slab ocean circulations to calculate and compare the equilibrium climate sensitivity.

Minor comments:

1. To distinguish model simulation from proxy reconstruction, I suggest the authors use the word "simulated", when talking about simulation results, e.g., Line 15.

2. Line 4–5 & Line 660: Simulations with $\sim 2°$ atmosphere and $\sim 1°$ ocean are not high resolution, but IPCC-like model resolution.

3. Line 89–95: Those are details of model setup, sort of redundant with descriptions around Line 115. Please consider moving details about model setup to Section 2.

4. Line 129–130: "global mean, volume weighted ocean temperature". Delete "average"

5. Line 205: In terms of an oxygen ...? CHANGE therms to terms

6. Line 240: ... ocean eddies (Viebahn et al., 2016).

7. Line 367: 30C –> 30°C

8. Line 369–371: "Still, most of the lower estimates are best matched by the annual mean SST in the 38Ma 4x PIC case while also meeting the higher estimates when considering summer maxima." Please rephrase.

9. Line 544: SST's –> SSTs?

---

## Referee Comment (RC2) · Anonymous Referee #2 · 4 Sep 2018

The paper of Baatsen et al. presents results from two simulations done using a recent version of the NCAR model (CMIP-5 like ?) and the late Eocene paleogeography they developed in an older paper. While I think these simulations worth to be published, an important work of re-writing is required before making this manuscript easy to follow and fluid.

This manuscript suffers from two issues; 1) many vague statements, story telling not always supported by the figures are found and 2) too many subjects are touched upon. I suppose that in their willingness to be sufficiently exhaustive, the authors have forgotten to keep focused their paper. I tried below to follow the paper and to suggest some corrections and/or some parts to be deleted.

I suggest to the authors to divide their manuscript into two papers. In the first one,

they could leave all parts on the general description of the results and the model-data comparison. In the second one, they can focus on the climatic sensitivity and the super-rotation. Following that recommandation, they may add more info to this part which is not alway very convincing. It is one possibility; they may find other means to arrange the manuscript.

In addition, while I do appreciate the work achieved by Baatsen et al. in reconstructing the late Eocene geography, they have to be more balanced and to recognize that many uncertainties exist and that other paleogeographies may be tested (see all papers by Ran Zhang, Zhongshi Zhang and its colleagues). Here, it looks like you have the ultimate paleogeography and all others are worst than yours; this is not the case and you should be more cautious. In addition, even if one accepts your paleogeography as being the most accurate; another uncertainty is the paleotopography. For example, topography over the Himalayan - Tibetan plateau area is badly constrained and may strongly impact both global climate and ocean dynamics (see paper in Climate of the Past by Su, B., Jiang, D., Zhang, R., Sepulchre, P., and Ramstein, G).

To conclude on the climate sensitivity. The model simulates a global warmth of 10°C from the CTRL run to the 2xPIC. Many factors are responsible for it; land-ice, vegetation cover and paleogeography. Without performing factorisation as already done by Lunt and colleagues, it remains hard to attribute a clear reason to what is occurring here. Conversely, the model simulate a 3°C warming between 4xPIC and 2xPIC, here the climate sensitivity is the classical one of the NCAR family model. So, I am left puzzled with the references to higher spatial resolution and to change in atmospheric grid to explain your results.

Specific/major Comments: l. 88 not entirely true, see Licht et al. (Nature, 2014). I do understand the need for clarity when writing a paper. In that sense, Baatsen et al. simplify the story and claim that no Late Eocene paleogeography have been used before them. But this is not true, Licht et al., in 2014, have also tested a Late Eocene paleogeography, not for the same goal but still . . . please modify the paper accordingly.

l. 94 I dont remember accurately conclusions reached by Beerling et al. in their 2011 PNAS paper but should be nice to state here the degree of global warming potentially reached when doubling and/or quadrupling methane concentration during the Eocene as it is what they did in their 2011 paper.

l. 110 - 90 x 144 correspond to 2° in latitude and 2.5° in longitude

l. 111 - avoid vague statement such as Âń´generally tropical . . . and mixed forests further polewardÃ㴴. Are you imposing zonal band of vegetation ? in addition, are you using phenological models of vegetation, i.e. interactive Leaf Area Index ? or is is an only PFT models by which the climate is influenced by biophysical parameters such as albedo, roughness and evapo-transpiration ?

Figure 1 - While the 2x simulation is close to equilibrium, the 4x would require more integration time before reaching the equilibrium as testified by Fig. 1b. So, lines 140-144 are misleading. One can also think that the reduced vertical mixing in the 4xPIC case is due to the still non equilibrated deep ocean. In addition, Figure 2b shows that age tracers are more equilibrated for the 4xPIC case than for the 2xPIC case. This is inverse to what is shown by Figure 1b. Why that ? Finally, the constant trend of THC index (l. 148) is hard to be interpreted because ocean dynamics can behave non linearly and can show tipping point after several hundred years of constancy. I like the way the authors do carefully treat the question of the equilibrium in their two simulations. It is something which matters a lot. However, putting more and more diagnostics can be also troubling ! So, my last comments are more questions open to every one in the modeling community than a direct criticism of the author's work.

Figure 3, this is an author choice but personally, I find the color palet very awkward and hard to look at. 3.1 If you describe your results, please dont mix with discussion ?

l. 170-175 you are referring to stratification but I do not see stratification indice on Figure 3 unless you are using the poorly visible contouring of the mixed layer depth ? This discussion is not supported by your model outputs and following lines 175-177

are even worst. I suggest removing all these lines from 170-177 and move them to a discussion part.

l.181 - path and depth (200-500 m) > are you meaning that the depth of the DP in your paleogeography is from 200 to 500 m ? Not clear

l. 185 where the reader should look at ?? to see the southward location of the polar front ??

l. 191-193 very vague statement that your results compare well with those of Hutchinson based on what ?

l. 178 - 193 - modeling results from this study + data + modeling from others are mixed here . . . once again, this is a mess to follow. Figure 3c, thick white lines show 0Pa - OK - but you also have a thick white line owing to the transition from negative to positive barotropic stream function. This is a bit harsh to follow. > l. 202 to l. 207 so what ?? can you discuss these results with those from organic data, is the amplitude the same ? etc . . .

l. 196-209 - Back to global results in term of SST and still with a mix of data and model.

l. 210-240 - Global overturning functions and associated OHT

l. 215-217 - So What ??

l. 221-222 - this sentence is not true at all. Figure 4d shows that more than a half of the OHT between the equator and 50°S (so not low latitudes but low to middle latitudes) are not due to the Pacific (compare red and black lines).

l. 226-230 - once again, these statements are not supported by your model results. From 50°S to south pole, all lines are converging. So, you can not write that the deep meridional overturning cell explains this feature in your paleo run as the OHT is the same in the modern while the overturning stream function is not ! l. 229-231 - So What ?

l. 243-259 Back to global feature and time evolution over the simulation Why here ??

l. 251-253 - Provide some details or just remove because it is a bit harsh written like that with no references.

l. 262-305 Data-Model comparisons - Can you provide an overall scoring using sum of square residuals for example ? it will be better than subjective term like mixed, substantial or good agreement. An other reason explaining data model disagreement may also be linked to the time period you are using. In 4 Myrs, CO2 and orbital parameters may have changed, hence explaining data spreading not represented by your modeling.

l. 271 Can you provide a reference for the saturation of UK37 above 28°C ?

l. 278-282 Remove these sentences, they bring nothing to the debate. As you are using a 4 per 4 ° box to generate temperature comparable to data; it is now hard to write that you have slight discrepancy in the position and/or model's limitation to resolve sharp upwelling zone (or at least for this point, you need to refer to previous studies showing that . . .). Once again, avoid vague statement and/or non supported statement. The next sentence is not understandable (the existence of colder temperatures in the . . .).

l. 305-330 Figure 3 but with winter/summer features. Suggest to remove all this part. 1) plotting the zonally average summer TOA incoming solar radiation, you will see that there is a decrease around 60° and then an increase again till the south Pole. So, no need to solve any mysteries concerning the minimum summer temperatures in the center of polar gyres (if they are centered around 65°). 2) all the discussion referring to previous work by Bijl and Huber is hard to follow without a sketch pointing out where are the currents, the authors are referring to. 3) nothing new in writing that winter hemisphere has a deeper thermal gradient and thus winds and gyres.

l. 332-371 Comparisons with some previous runs. Suggest to remove all this part. Once again, many crucial numbers and references are lacking in this part making it vague and not very convincing. Can you state reference's papers for each version of

the NCAR model you are referring to ? you first write that the spatial resolution is not the same, and then, the initial conditions and then the numerical grid in the atmosphere and also the aerosols. What about the duration of each run ? And after all, why doing this comparison if there are so many things that have evolved since the GH14 paper . . . Figure 8 may be interpreted as showing two very different states of the climate and so what's the point here ? two very different version of the same model shows two very different climatic state ? Figure A3 also shows that the old version was simulating a latitudinal thermal gradient more flat than the new version (i.e. compare dashed red line and green line).

l. 340-342 Can you provide a reference for such a statement ? l. 342-344 Conversely, comparison with results from Hutchinson is light whereas the paleogeography is the same. For example, you write Âǎsimilar global temperatures are found . . .Âǎ(l. 343-344). Can you give numbers mean global SST and air temperature for both models, yours and the one of Hutchinson ? l.346-359 This is not the same geography and the CO2's sensitivity is not the same. So, no surprises, results are different !

l. 374-400 General Description of the atmosphere. l. 375-376 - Where are results from 2xPIC in this part ? l. 403-426 General Description of the zonally averaged distribution of 2 m air temperature with 2x and 4x. Here many classical results are found by the authors . . . in particular about the polar amplification phenomenon. I am not sure that it brings anything new to the knowledge of the paleoclimate community. Could be remove to gain place and clarity.

l. 429-467 Data model comparison over land

l. 471-472 So, you are not able to resolve data model disagreement as many authors suggest winter positive temperature in the continental interior of Siberia (see work of Upchurch, Valdes, Suan etc . . .) l. 476-478 are you using interactive phenology in your set-up ? in this case, you should be able to show the LAI or the vegetation cover ? if not, your explanation is not possible. If yes, please show the simulated distribution

vegetation cover (not the imposed one). If you impose a forest in an arid area, LAI should be 0 and the vegetation model should be able to predict no forests and you should have only barren soil. l. 495-525 Zonal winds and super rotation . . .

l.532 can you provide main principles of the Etminan method ? as the radiative forcings is a logarithmic function of greenhouse gas concentration, it is generally admitted that each $CO_2$ doubling will have the same influence (if not accounting for internal feedbacks), here you have 4.23 and 4.48, is the difference due to internal feedbacks of the Earth System Model ? l.540-544 not sure to get your point here. If you are in a virtually ice-free world, how can, you have albedo effects due to snow (add seasonal polar view of snow and sea ice cover) ? is it more related to clouds feedbacks for example ? Again, concerning the growing season, this should be demonstrated and not just written. Or at least, expand your discussion here. Otherwise, it looks again as vague statement.

l. 545-547 is it due to larger snow and sea-ice cover in our modern climate ? l. 558-565 Interpreting OLR is always ambiguous as the thermal response does influence it. Here, you write that you will explain why you have a polar amplification but you end up by writing that you have more OLR because your high latitudes are warmer. No explanations of why they are warmer ? Or I have missed something ?

In addition, I keep in mind from the Caballero and Huber PNAS paper that the super rotation strongly decrease the low cloud cover, hence leaving more solar radiation penetrating the atmosphere and warming it. Do you see that between 4X and 2X ?

Fig.14 f : Could cover, changes between -5 and + 5 %, looks very small and the same range than above (14d) ? Otherwise, it confirms what I suspect, i.e. more high clouds in your mid-to-high latitudes that may induce warmer high latitudes ? This is however possibly offset by changes in low cloud cover . . .

l. 586-600 - not clear, please provide more details. Your first $CO_2$ doubling (l. 590) has a larger effect because you also do account for changes in vegetation cover, land-ice,

paleogeography etc ... Then, you write that we can not add the radiative forcing but the, you do this addition assuming a linear behavior ... How did you get ESS = 5.68-8.97 ?? and 6°C per CO2 doubling ? (you have 3 °C in the Table 3 from 2xPIC to 4xPIC, or you have 10 °C for the first round but accounting for all geological factors)

l.635-652 - Please provide more details or at least, explain where you want to go with all these numbers.

Summary and conclusions:

l. 685-690 Once again, these statements have not been demonstrated in this paper. Higher resolution is not the solution to everything and in case, you have to explain which physical processes are impacted by a higher resolution and how they explain your warmer climate simulation. The statement on the atmospheric grid is also doubtful - this is maybe true but please, provide explanations, what changes are occurring when modifying the grid. Why your more specific geography reconstruction should allow for a better representation of the Antarctic summer warmth and zonal heterogeneity in the souther ocean. To conclude, I am still left with many unanswered question, the main one is why you have this 10°C warming between your control run and between you 2xPIC ? it is probably due to a mixed of land-ice, vegetation changes and paleogeography effect. However, to provide a firm quantification of the effect of these boundary conditions, you have to do the factorisation method as done by Dan and others in the Miocene I guess. Conversely, your CO2 model sensitivity remain in the lower range with 3°C (from 2xPIC to 4xPIC). Hence, I do not understand why you want to incriminate higher resolution and/or changes in atmospheric grids to explain your global warmth. You first have a large change in temperature when comparing CTRL to 2xPIC but then when comparing 4xPIC to 2xPIC, you have a more classical answer.

---

## Author Comment (AC1) · 2 Oct 2018

We would like to thank the reviewer for the detailed review and comments regarding this paper. The main issues concern the clarity (both in text structure and language) and length of the manuscript. To make an improvement on both points, we will re-write and re-structure the paper in order to make it more focussed. There will be more attention to the model set-up, while the general description and discussion of the results will be more limited. Several figures regarding the spin-up and general circulation will be taken out, to focus on model-proxy and model-model comparisons. The discussion on climate sensitivity will be simplified as well. This will result in an overall shorter, more focussed and better structured paper to meet most of the reviewer's requests.

A detailed, point-by-point response is given in the added file AR1.pdf

Please also note the supplement to this comment:
https://www.clim-past-discuss.net/cp-2018-43/cp-2018-43-AC1-supplement.pdf

————————————————————

[Figure]

**Supplement:**

**Author's response to Referee #1**

**Referee Comment (RC)**: This manuscript presented simulations of the middle-to-late Eocene climate using the CESM1.0. The model resolution was 1-2°, higher than most of the previous Eocene simulations. A new set of geographical boundary conditions was employed. The authors claimed that their modelling results were in good agreement with proxy records. They also described many aspects of ocean/atmosphere states and circulations and climate sensitivity in the simulations.

Simulations in the study could potentially contribute to our understanding of the past warm worlds, but I do not find this manuscript in the current form satisfactory. To make this manuscript more accessible to readers, the writing should be improved greatly. Vague expressions should be avoided. To substantiate many of the authors' arguments, new analyses need to be conducted. Please see detailed comments below.

**Author's Response (AR)**: *We would like to thank the reviewer for the detailed review and comments regarding this paper. The main issues concern the clarity (both in text structure and language) and length of the manuscript.*
*To make an improvement on both points, we will re-write and re-structure*
*the paper in order to make it more focussed. There will be more attention to the model set-up, while the general description and discussion of the results will be more limited. Several figures regarding the spin-up and general circulation will be taken out, to focus on model-proxy and model-model comparisons. The discussion on climate sensitivity will be simplified as well.  This will result in an overall shorter, more focussed and better structured paper to meet most of the reviewer's requests.*

**RC**: Major comments:
**1**. The authors used many vague expressions in their manuscript. I only list a few here for illustrative purpose. I suggest the authors carefully go through the manuscript and improve the writing.
Line 285: ". . . in good agreement with the model."
Line 286–287: "A mixed agreement is seen at other latitudes, with model results being too warm in the northern middle latitudes, too cold in southern middle latitudes and good at high latitudes."
Line 300: "Good agreement between proxy and model results . . ."
Line 442: ". .agree fairly well . . ."
Line 674: ". . . makes a fairly good match . . ."

**AR**: *These parts will be re-written, avoiding vague expressions and clarifying the statements made where needed.*

**2**. This is a long manuscript with 18 items (figures + tables). I would suggest the authors simplify the figures, shorten the results but highlight and provide in-depth analyses on the differences from previous modeling results. For example, in Figure 1-2, it is not very interesting to show the time series of individual basins (Pacific and Atlantic). In fact, this detailed information is not much discussed in the manuscript. I suggest the authors focus more on the relevant findings and major difference from previous simulations, to make this manuscript more interesting to read.

**AR**: *Some of the general modelling results will be taken out and figures will be simplified as suggested. This will leave room for more discussion on model-model comparisons.*

**3**. The authors need to do a better job describing their model and experimental setup, to ensure that results are reproducible by others. For example, CESM1.0.5 has active land and sea ice model, which the authors did not mention in the manuscript. Also, what values are used for the orbital parameters, solar constant, and other greenhouse gases (N2O and CFCs)? How is the land surface configured? How are the lakes and rivers treated? What is the soil color in the Eocene simulations? How is the ocean mixing (including tidal dissipation) parameterized? I know this list is long, but it is essential for other researchers to reproduce and correctly interpret results in this study.

**AR**: *The model description will be extended in view of these comments. The treatment of the land surface and sea ice in the model are simplified, but will nonetheless be presented and motivated at the start of the paper.*

**4**. I find results on model-data comparison and analyses on climate sensitivity more interesting and relevant than descriptions of many detailed aspects of atmospheric and oceanic state/circulations. I think, when describing the modeling results, after a brief discussion of the model spin-up process, it is better to show modelled results compared with proxy records first. This essentially gives readers a general ideal about the model performance and its advantage/disadvantage compared with previous modeling results.

**AR**: *In light of this comment, the description of general model results will be reduced while the parts on climate sensitivity and proxy/model comparisons will be re-worked.*

**5**. Aside from the analyses on climate sensitivity, this manuscript is highly descriptive with many statements requiring more in-depth analysis to substantiate.
Example 1: when explaining the temperature difference between the present study and Goldner et al. (2014), the authors listed a few possible factors (e.g., model resolution, dynamic core, radiative forcing from CH4, aerosol, . . .) but fail to, at least try to, quantify contribution from any factor.

**AR**: *We will provide a more in-depth discussion on what causes the significant differences to other simulations using a similar model. More  elaborate comparisons will be made in order to quantify some of the specific contributions.*

Example 2: the authors ascribed the different climate sensitivity between the Eocene climate and the present day to "fast feedback processes". They failed to explain further what these fast feedback processes are? Are there other processes involved, like clouds, albedo and lapse rate? Have you quantified the feedback strength (e.g., using the partial radiative perturbation) to make the argument?

**AR**: *The main point made in the original manuscript is that conventional instant doubling or quadrupling of $CO_2$ experiments never show the true equilibrium climate sensitivity, but only that of fast feedback processes. What is fast depends on the specific experiment. In our paper, two separately equilibrated simulations are presented, providing a different framework to determine climate sensitivity. We will make this more clear in the revised paper.*

Example 3: Around Line 22–23, the authors stated that ". . . without the need for greenhouse gas concentrations much higher than proxy estimates". What are the estimated ranges of CO2 and CH4 from reconstructions? How different are these reconstructions compared with values used in simulations in this study?
There are many other unsubstantiated statements all over the manuscript.

**AR**: *There is quite a wide range of proxy estimates for the middle-to-late Eocene, mostly between 500-1500ppm $CO_2$ (Anagnostou et al. 2016). The simulations here show a good match with proxies under 560/1120ppm (and $CH_4$ doubling/quadrupling), which falls well within the possible range. This will be presented in more detail in the revised paper.*

**6**. Using simplified equations in Etminan et al. (2016) for radiative forcing calculation in a 3D climate model is not justified. For example, the adjusted radiative forcing for a CO2 doubling at preindustrial is 3.5 W m-2 from off-line radiation calculation (Bitz et al., 2012), rather than 3.8 W m-2 from equations in Etminan et al. (2016). It is not clear how much the difference could be under high CO2 and CH4 levels. The authors need either to calculate the radiative forcing from the off-line version of CESM radiation code, or to conduct slab ocean circulations to calculate and compare the equilibrium climate sensitivity.

**AR**: *Additional calculations will be carried out to determine the exact radiative forcing perturbations in the model and to get a more accurate estimate of climate sensitivity.*

**RC**: Minor comments:
**1**. To distinguish model simulation from proxy reconstruction, I suggest the authors use the word "simulated", when talking about simulation results, e.g., Line 15.

**AR**: *This will be taken into account to better separate the two.*

**2**. Line 4–5 & Line 660: Simulations with 2° atmosphere and 1° ocean are not high resolution, but IPCC-like model resolution.

**AR**: *True, they are relatively high resolution for palaeo standards but not actually high. These statements will be adjusted accordingly.*

**3**. Line 89–95: Those are details of model setup, sort of redundant with descriptions around Line 115. Please consider moving details about model setup to Section 2.

**AR**: *Descriptions regarding model set-up will be moved into the (extended) section on the model configuration.*

**4**. Line 129–130: "global mean, volume weighted ocean temperature". Delete "average"

**AR**: *This will be adjusted to avoid a double statement*.

**5**. Line 205: In terms of an oxygen . . .? CHANGE therms to terms 6.
Line 240: . . . ocean eddies (Viebahn et al., 2016).

**AR**: *These will be corrected.*

**7**. Line 367: 30C –> 30°C

**AR**: *Will be corrected and checked throughout.*

**8**. Line 369–371: "Still, most of the lower estimates are best matched by the annual mean SST in the 38Ma 4x PIC case while also meeting the higher estimates when considering summer maxima." Please rephrase.

**AR**: *This sentence will be split and re-written.*

**9**. Line 544: SST's –> SSTs?

**AR**: *Will be changed.*

---

## Author Comment (AC2) · 2 Oct 2018

We would like to thank the reviewer for the extensive and in depth comments of the manuscript. We agree to shorten the paper mainly in terms of figures/tables and cutting back on the description of model. The structure and language of the manuscript will both be improved to increase readability and clarity.

For a detailed, point-by-point reply please see the added file AR2.pdf

Please also note the supplement to this comment:
https://www.clim-past-discuss.net/cp-2018-43/cp-2018-43-AC2-supplement.pdf

[Figure]

**Supplement:**

**Author's response to Referee #2**

**Referee Comment (RC)**: The paper of Baatsen et al. presents results from two simulations done using a recent version of the NCAR model (CMIP-5 like ?) and the late Eocene paleogeography they developed in an older paper. While I think these simulations worth to be published, an important work of re-writing is required before making this manuscript easy to follow and fluid.

This manuscript suffers from two issues; 1) many vague statements, story telling not always supported by the figures are found and 2) too many subjects are touched upon. I suppose that in their willingness to be sufficiently exhaustive, the authors have forgotten to keep focused their paper. I tried below to follow the paper and to suggest some corrections and/or some parts to be deleted.

I suggest to the authors to divide their manuscript into two papers. In the first one, they could leave all parts on the general description of the results and the model-data comparison. In the second one, they can focus on the climatic sensitivity and the super-rotation. Following that recommandation, they may add more info to this part which is not alway very convincing. It is one possibility; they may find other means to arrange the manuscript.

**Author's Response (AR)**: *We would like to thank the reviewer for the extensive and in depth comments of the manuscript. We agree to shorten the paper mainly in terms of figures/tables and cutting back on the description of model. The structure and language of the manuscript will both be improved to increase readability and clarity.*

**RC**: In addition, while I do appreciate the work achieved by Baatsen et al. in reconstructing the late Eocene geography, they have to be more balanced and to recognize that many uncertainties exist and that other paleogeographies may be tested (see all papers by Ran Zhang, Zhongshi Zhang and its colleagues). Here, it looks like you have the ultimate paleogeography and all others are worst than yours; this is not the case and you should be more cautious. In addition, even if one accepts your paleogeography as being the most accurate; another uncertainty is the paleotopography. For example, topography over the Himalayan - Tibetan plateau area is badly constrained and may strongly impact both global climate and ocean dynamics (see paper in Climate of the Past by Su, B., Jiang, D., Zhang, R., Sepulchre, P., and Ramstein, G).

**AR**: *The aim here is to show that with increased model resolution and the most recent information on palaeogeography, a better representation of the middle-to-late Eocene climate can be achieved. It is certainly not the intention to claim that the model set-up and reconstruction are perfect, so this will be better nuanced. Since the geography reconstruction was made based on a Paleomag rather than Hot spot plate-tectonic reference, it also serves to show the array of possible uncertainty regarding geographical boundary conditions rather than claiming that one is better.*

**RC**: To conclude on the climate sensitivity. The model simulates a global warmth of $10{\circ}C$ from the CTRL run to the 2xPIC. Many factors are responsible for it; land-ice, vegetation cover and paleogeography. Without performing factorisation as already done by Lunt and colleagues, it remains hard to attribute a clear reason to what is occurring here. Conversely,

the model simulate a 3∘C warming between 4xPIC and 2xPIC, here the climate sensitivity is the classical one of the NCAR family model. So, I am left puzzled with the references to higher spatial resolution and to change in atmospheric grid to explain your results.

**AR**: *Both the resolution and atmospheric grid used here were previously shown to increase the sensitivity of the CESM model. Clearly the model is also experiencing a significant heating from the Eocene boundary conditions, which is more substantial than in most previous studies but similar to the ~5 C of Caballero and Huber (2013). This will be discussed in more detail and additional analysis regarding the sensitivity and temperature offset will be carried out.*

**RC**: Specific/major Comments: l. 88 not entirely true, see Licht et al. (Nature, 2014). I do understand the need for clarity when writing a paper. In that sense, Baatsen et al. simplify the story and claim that no Late Eocene paleogeography have been used before them. But this is not true, Licht et al., in 2014, have also tested a Late Eocene paleogeography, not for the same goal but still . . . please modify the paper accordingly.

**AR**: *Thanks for pointing out this shortcoming. A 40Ma reconstruction was indeed used by Licht et al. but there is little information on how it was created. They also use a coarse resolution coupled model (FOAM) to drive an atmosphere-only simulation, which is not really comparable to the simulations presented here.*

**RC**: l. 94 I dont remember accurately conclusions reached by Beerling et al. in their 2011 PNAS paper but should be nice to state here the degree of global warming potentially reached when doubling and/or quadrupling methane concentration during the Eocene as it is what they did in their 2011 paper.

**AR**: *According to Etminan et al. (2016), the contribution of $CH_4$ is limited and makes the radiative forcing equivalent to that of 2.15x and 4.69x PI $CO_2$, respectively, as mentioned later in the paper. This will be mentioned now earlier in the paper for clarity.*

**RC**: *l. 110 - 90 x 144 correspond to 2∘ in latitude and 2.5∘ in longitude.*

**AR**: *This was a typo which will be corrected; the model grid should be 96x144 which does correspond to 1.9x2.5deg*

**RC**: l. 111 - avoid vague statement such as 'generally tropical ... and mixed forests further poleward'. Are you imposing zonal band of vegetation? in addition, are you using phenological models of vegetation, i.e. interactive Leaf Area Index? or is is an only PFT models by which the climate is influenced by biophysical parameters such as albedo, roughness and evapo-transpiration?

**AR**: *The implementation of vegetation is rather simple and consists of static PFT's determining their response in the model. This should be treated in more detail within the model set-up and maybe demands a figure showing model geography and vegetation.*

**RC**: Figure 1 - While the 2x simulation is close to equilibrium, the 4x would require more integration time before reaching the equilibrium as testified by Fig. 1b. So, lines 140-144 are misleading. One can also think that the reduced vertical mixing in the 4xPIC case is due to the still non equilibrated deep ocean. In addition, Figure 2b shows that age tracers are more equilibrated for the 4xPIC case than for the 2xPIC case. This is inverse to what is shown by Figure 1b. Why that ? Finally, the constant trend of THC index (l. 148) is hard to be interpreted because ocean dynamics can behave non linearly and can show tipping point after several hundred years of constancy. I like the way the authors do carefully treat the question of the equilibrium in their two simulations. It is something which matters a lot. However, putting more and more diagnostics can be also troubling ! So, my last comments are more questions open to every one in the modeling community than a direct criticism of the author's work.

**AR**: *The 4x PIC is indeed showing larger trends than the 2x PIC one, but the scaling in the figure make those appear quite dramatic. Looking at the values in Table 2, there is about 0.5C warming per 1000 years which is quite small and in fact similar or less than what is seen in most other studies (e.g. 0.016C/century in Goldner et al. 2014). It should be acknowledged that the model is not yet fully equilibrated here, but to get rid of this trend would require several 1000s more years of model integration. The time series of MOC strength show that there is no trend over the last 1000 model years, so there seems to be no link to slowly warming deep ocean temperatures. The different trends in age tracers is caused by the deep mixing of younger waters opposed to ageing of stagnant regions; this will be mentioned in the revised paper.*

**RC**: Figure 3, this is an author choice but personally, I find the color palet very awkward and hard to look at. 3.1 If you describe your results, please dont mix with discussion?

**AR**: *The colour scales were designed specifically to be easy to interpret and friendly to people with mild colour blindness (they were also used in Baatsen et al. 2018). Maybe an adjustment of the scaling and density of contour lines can help improve the readability. This section will be restructured to have only a short presentation of general results and a discussion later on.*

**RC**: l. 170-175 you are referring to stratification but I do not see stratification indice on Figure 3 unless you are using the poorly visible contouring of the mixed layer depth? This discussion is not supported by your model outputs and following lines 175-177 are even worst. I suggest removing all these lines from 170-177 and move them to a discussion part.

**AR**: *Low stratification was linked to high upper level densities; this will be clarified. The discussion will also be moved further down.*

**RC**: l.181 - path and depth (200-500 m) > are you meaning that the depth of the DP in your paleogeography is from 200 to 500 m ? Not clear

**AR**: *This is regarding the model geography, adding a figure on this should help.*

**RC**: l. 185 where the reader should look at ?? to see the southward location of the polar front ??

**AR**: *The front is best seen in the barotropic stream function; this will be mentioned.*

**RC**: l. 191-193 very vague statement that your results compare well with those of Hutchinson based on what?

**AR**: *A figure will be added with a comparison to proxies and other model studies (with a description of the sources of the data).*

**RC**: l. 178 - 193 - modeling results from this study + data + modeling from others are mixed here . . . once again, this is a mess to follow.
Figure 3c, thick white lines show 0Pa – OK - but you also have a thick white line owing to the transition from negative to positive barotropic stream function. This is a bit harsh to follow.
> l. 202 to l. 207 so what ?? can you discuss these results with those from organic data, is the amplitude the same ? etc . . .

**AR**: *Most of this part will be moved to the discussion section; the figure will be improved. The motivation to use 2x and 4x PIC forcing is to cover middle-to-late Eocene temperatures so it is relevant to mention a similar range is seen in estimates from benthic $d^{18}O$.*

**RC**: l. 196-209 - Back to global results in term of SST and still with a mix of data and model.
l. 210-240 - Global overturning functions and associated OHT
l. 215-217 - So What ??

**AR**: *This part will be re-written.*

**RC**: l. 221-222 - this sentence is not true at all. Figure 4d shows that more than a half of the OHT between the equator and 50◦S (so not low latitudes but low to middle latitudes) are not due to the Pacific (compare red and black lines).

**AR**: *This will be adjusted to show Indo-Pacific rather than Pacific-only similar to the Pre-industrial case. The description will be re-phrased accordingly.*

**RC**: l. 226-230 - once again, these statements are not supported by your model results. From 50◦S to south pole, all lines are converging. So, you can not write that the deep meridional overturning cell explains this feature in your paleo run as the OHT is the same in the modern while the overturning stream function is not !
l. 229-231 - So What ?

**AR**: *The difference related to Ekman upwelling in the ACC is quite clear, but the description should be modified as there are indeed no changes beyond ~55S. The main point remains that a significant redistribution of heat fluxes takes place from the NH towards the SH.*

**RC**: l. 243-259 Back to global feature and time evolution over the simulation Why here ??
l. 251-253 - Provide some details or just remove because it is a bit harsh written like that with no references.

**AR**: *These parts will be re-arranged.*

**RC**: l. 262-305 Data-Model comparisons - Can you provide an overall scoring using sum of square residuals for example ? it will be better than subjective term like mixed, substantial or good agreement. Another reason explaining data model disagreement may also be linked to the time period you are using. In 4 Myrs, CO2 and orbital parameters may have changed, hence explaining data spreading not represented by your modeling.

**AR**: *The scarcity and quality of the available proxies do not allow the use of objective statistical analyses or shorter time intervals here. Proxies can be strongly biased or a slight shift in strongly regionally-dependent conditions can easily mess up such a comparison. The most transparent way is to just show the proxies with model results and leave to the reader the interpretation of individual points.*

**RC**: l. 271 Can you provide a reference for the saturation of UK37 above 28∘C ?

**AR**: *Yes, e.g. Conte (2006), will be added.*

**RC**: l. 278-282 Remove these sentences, they bring nothing to the debate. As you are using a 4 per 4 ∘ box to generate temperature comparable to data; it is now hard to write that you have slight discrepancy in the position and/or model's limitation to resolve sharp upwelling zone (or at least for this point, you need to refer to previous studies showing that ...). Once again, avoid vague statement and/or non supported statement. The next sentence is not understandable (the existence of colder temperatures in the . . .).

**AR**: *This part will be rephrased. The model spatial variation overlaps with that of the proxies, so it can actually motivate the presence of nearby cooler waters.*

**RC**: l. 305-330 Figure 3 but with winter/summer features. Suggest to remove all this part.
**1**) plotting the zonally average summer TOA incoming solar radiation, you will see that there is a decrease around 60∘ and then an increase again till the south Pole. So, no need to solve any mysteries concerning the minimum summer temperatures in the center of polar gyres (if they are centered around 65∘).
**2**) all the discussion referring to previous work by Bijl and Huber is hard to follow without a sketch pointing out where are the currents, the authors are referring to.
**3**) nothing new in writing that winter hemisphere has a deeper thermal gradient and thus winds and gyres.

**AR**: *As suggested, this figure will be removed along with most of the discussion.*

**RC**: l. 332-371 Comparisons with some previous runs. Suggest to remove all this part. Once again, many crucial numbers and references are lacking in this part making it vague and not very convincing. Can you state reference's papers for each version of the NCAR model you

are referring to ? you first write that the spatial resolution is not the same, and then, the initial conditions and then the numerical grid in the atmosphere and also the aerosols. What about the duration of each run ? And after all, why doing this comparison if there are so many things that have evolved since the GH14 paper . . . Figure 8 may be interpreted as showing two very different states of the climate and so what's the point here ? two very different version of the same model shows two very different climatic state ? Figure A3 also shows that the old version was simulating a latitudinal thermal gradient more flat than the new version (i.e. compare dashed red line and green line).

**AR**: *This figure will be removed and replaced by a more general overview of model results and proxies. The aim of this figure was to show the improvement being made in the general output and the added detail in the results, but it does not add much in general. There is only a slightly reduced thermal gradient in the Southern Hemisphere, which is the result of a weaker seasonal cycle (Figure A4) and the use of a completely closed Tasmanian Gateway in GH14.*

**RC**: l. 340-342 Can you provide a reference for such a statement ?

**AR**: *This is an observation from running the model in different configurations; there are no publications that make a 1-on-1 comparison under similar conditions.*

**RC**: l. 342-344 Conversely, comparison with results from Hutchinson is light whereas the paleogeography is the same. For example, you write 'similar global temperatures are found ...'(l. 343- 344). Can you give numbers mean global SST and air temperature for both models, yours and the one of Hutchinson ?
l.346-359 This is not the same geography and the $CO_2$'s sensitivity is not the same. So, no surprises, results are different !

**AR**: *As suggested above, this discussion will be removed and a better comparison will be given with the results of Hutchinson et al. (2018) in a new figure.*

**RC:** l. 374-400 General Description of the atmosphere.
l. 375-376 - Where are results from 2xPIC in this part ?

**AR**: *As mentioned, they are qualitatively similar and therefore only regarded in zonal mean plots and tables.*

**RC**: l. 403-426 General Description of the zonally averaged distribution of 2 m air temperature with 2x and 4x. Here many classical results are found by the authors . . . in particular about the polar amplification phenomenon. I am not sure that it brings anything new to the knowledge of the paleoclimate community. Could be remove to gain place and clarity.

**AR**: *The figure and description of zonal mean air temperatures will be removed and solely discussed in the comparison to proxies.*

**RC**: l. 429-467 Data model comparison over land

l. 471-472 So, you are not able to resolve data model disagreement as many authors suggest winter positive temperature in the continental interior of Siberia (see work of Upchurch, Valdes, Suan etc . . .)

l. 476-478 are you using interactive phenology in your set-up ? in this case, you should be able to show the LAI or the vegetation cover ? if not, your explanation is not possible. If yes, please show the simulated distribution vegetation cover (not the imposed one). If you impose a forest in an arid area, LAI should be 0 and the vegetation model should be able to predict no forests and you should have only barren soil.

**AR**: *The PFTs are prescribed, but how they respond is dependent on the climatic conditions. Most of the papers on the interior of Siberia seem to consider Early Eocene or late Cretaceous? There will be some more discussion on this issue in the revised paper.*

**RC**: l. 495-525 Zonal winds and super rotation . . .

**AR**: *The figure and description of the atmospheric zonal mean vertical cross sections will be taken out.*

**RC**: l.532 can you provide main principles of the Etminan method ? as the radiative forcings is a logarithmic function of greenhouse gas concentration, it is generally admitted that each CO2 doubling will have the same influence (if not accounting for internal feed- backs), here you have 4.23 and 4.48, is the difference due to internal feedbacks of the Earth System Model?

**AR**: *The main reason for the nonlinearity is the interaction between different greenhouse gases; also the pure logarithmic behaviour is only the first-order contribution.*

**RC**: l.540-544 not sure to get your point here. If you are in a virtually ice-free world, how can, you have albedo effects due to snow (add seasonal polar view of snow and sea ice cover) ? is it more related to clouds feedbacks for example ? Again, concerning the growing season, this should be demonstrated and not just written. Or at least, expand your discussion here. Otherwise, it looks again as vague statement.

**AR**: *There is very little snow/ice, especially at 4x PIC but there is still winter snow cover. At 2x PIC there is still limited snow/ice cover but there is some influence on the climatic response that will  be mentioned. The effect of vegetation and its response will be looked at in more detail.*

**RC**: l. 545-547 is it due to larger snow and sea-ice cover in our modern climate ?

**AR**: *Mainly the difference in snow/ice is thought to be the cause, but this is not discussed as there is no 2x PIC pre-industrial control experiment.*

**RC**: l. 558- 565 Interpreting OLR is always ambiguous as the thermal response does influence it. Here, you write that you will explain why you have a polar amplification but you end up by writing that you have more OLR because your high latitudes are warmer. No explanations of why they are warmer ? Or I have missed something ?

**AR**: *The OLR (clearsky) mainly shows the response to higher temperatures, but the main contribution is from clouds. This will be described more clearly.*

**RC**: In addition, I keep in mind from the Caballero and Huber PNAS paper that the super rotation strongly decrease the low cloud cover, hence leaving more solar radiation penetrating the atmosphere and warming it. Do you see that between 4X and 2X ?

**AR**: *As shown in figure 14, there are some reorganisations but no clear reduction in cloud cover between 2x and 4x PIC so this does not seem to be the case here.*

**RC**: Fig.14 f : Could cover, changes between -5 and + 5 %, looks very small and the same range than above (14d) ? Otherwise, it confirms what I suspect, i.e. more high clouds in your mid-to-high latitudes that may induce warmer high latitudes ? This is however possibly offset by changes in low cloud cover . . .

**AR**: *The changes in zonal mean cloud cover are small but significant as shown by the TOM fluxes. Indeed, increased high cloud cover is keeping middle/high latitudes warm as is reduced low cloud cover over the Arctic in summer.*

**RC**: l. 586-600 - not clear, please provide more details. Your first $CO_2$ doubling (l. 590) has a larger effect because you also do account for changes in vegetation cover, land-ice, paleogeography etc . . . Then, you write that we can not add the radiative forcing but the, you do this addition assuming a linear behavior . . . How did you get ESS = 5.68- 8.97 ?? and 6∘C per $CO_2$ doubling ? (you have 3 ∘C in the Table 3 from 2xPIC to 4xPIC, or you have 10 ∘C for the first round but accounting for all geological factors)

**AR**: *The discussion regarding ESS/ECS and geography influences will be re-written accordingly.*

**RC**: l.635-652 - Please provide more details or at least, explain where you want to go with all these numbers.

**AR**: *Mainly this shows that the general results regarding S/G do not change using various methods, but it will be further clarified.*

**RC**: Summary and conclusions:

l. 685-690 Once again, these statements have not been demonstrated in this paper. Higher resolution is not the solution to everything and in case, you have to explain which physical processes are impacted by a higher resolution and how they explain your warmer climate simulation. The statement on the atmospheric grid is also doubtful - this is maybe true but please, provide explanations, what changes are occurring when modifying the grid. Why your more specific geography reconstruction should allow for a better representation of the Antarctic summer warmth and zonal heterogeneity in the souther ocean.

**AR**: *The higher resolution and new geography reconstruction mainly allows for a more regional interpretation of proxies to cause improved agreement on one hand and explain some discrepancies that remain. The effects of the grid and dynamical core will be discussed in more detail and further clarified.*

**RC**: To conclude, I am still left with many unanswered question, the main one is why you have this 10∘C warming between your control run and between you 2xPIC ? it is probably due to a mixed of land-ice, vegetation changes and paleogeography effect. However, to provide a firm quantification of the effect of these boundary conditions, you have to do the factorisation method as done by Dan and others in the Miocene I guess. Conversely, your $CO_2$ model sensitivity remain in the lower range with 3∘C (from 2xPIC to 4xPIC). Hence, I do not understand why you want to incriminate higher resolution and/or changes in atmospheric grids to explain your global warmth. You first have a large change in temperature when comparing CTRL to 2xPIC but then when comparing 4xPIC to 2xPIC, you have a more classical answer.

**AR**: *As stated, the model in its configuration used here responds strongly to the applied changes in boundary conditions but less so to increased radiative forcing from greenhouse gases. This will be looked at in more detail and discussed in the revised paper.*